# Local auxin biosynthesis acts downstream of brassinosteroids to trigger root foraging for nitrogen

Zhongtao Jia [1], Ricardo F. H. Giehl [1] & Nicolaus von Wirén [1✉]

Lateral roots (LRs) dominate the overall root surface of adult plants and are crucial for soil exploration and nutrient acquisition. When grown under mild nitrogen (N) deficiency, flowering plants develop longer LRs to enhance nutrient acquisition. This response is partly mediated by brassinosteroids (BR) and yet unknown mechanisms. Here, we show that local auxin biosynthesis modulates LR elongation while allelic coding variants of YUCCA8 determine the extent of elongation under N deficiency. By up-regulating the expression of *YUCCA8/3/5/7* and of *Tryptophan Aminotransferase of Arabidopsis 1* (*TAA1*) under mild N deficiency auxin accumulation increases in LR tips. We further demonstrate that N-dependent auxin biosynthesis in LRs acts epistatic to and downstream of a canonical BR signaling cascade. The uncovered BR-auxin hormonal module and its allelic variants emphasize the importance of fine-tuning hormonal crosstalk to boost adaptive root responses to N availability and offer a path to improve soil exploration by expanded root systems in plants.

[1] Molecular Plant Nutrition, Dept. Physiology and Cell Biology, Leibniz Institute of Plant Genetics and Crop Plant Research, Stadt Seeland, OT Gatersleben, Germany. ✉email: vonwiren@ipk-gatersleben.de

The root system of dicots is formed by one embryonically formed primary root and post-embryonically developed lateral roots (LRs) of different orders. The formation of LRs determines the horizontal expansion of a root system and the soil volume that can be exploited for nutrients and water. LR development commences with the specification of a group of xylem-pole pericycle cells in the basal meristem and continues with a series of tightly coordinated cell divisions to give rise to a dome-shaped LR primordium[1,2]. These steps are followed by the formation of a radially symmetrical LR meristem, which eventually penetrates the outer cell layers of the parental root and emerges to form a mature LR[1,2]. The development of LRs is highly plastic, responding with altered number, angle, and length to external nutrient availability and overall plant demand for nutrients[3–6].

Previous studies have revealed that N availability interferes with almost every checkpoint of LR development through recruitment of mobile peptides or by activating auxin signaling and other hormonal crosstalks[7–13]. If N in the form of nitrate is accessible only to a part of the root system, LRs elongate into the nitrate-containing patch under control of the auxin-regulated transcription factor ARABIDOPSIS NITRATE REGULATED 1 (ANR1)[14,15]. In contrast, local supply of ammonium triggers LR emergence by enhancing radial diffusion of auxin in a pH-dependent manner[16,17]. These developmental processes cease when plants are exposed to severe N limitation, which forces roots to adopt a survival strategy by suppressing LR development[11,18]. Suppression of LR outgrowth by extremely low N availability involves NRT1.1/NPF6.3-mediated auxin transport and the CLE-CLAVATA1 peptide-receptor signaling module[11,12,19]. Furthermore, LR growth under N-free conditions is controlled by the MADS-box transcription factor AGL21[20]. Notably, external N levels that provoke only mild N deficiency, common in natural environments or low-input farming systems, induce a systemic N foraging response characterized by enhanced elongation of roots of all orders[18,21–23]. Recently, we discovered that brassinosteroid (BR) biosynthesis and signaling are required for N-dependent root elongation[24,25]. Although the elongation of both the primary root (PR) and LRs are induced by mild N deficiency, LRs respond differentially to BR signaling. While PR and LR responses to low N were in overall similarly attenuated in BR-deficient mutants of Arabidopsis thaliana, loss of BRASSINOSTEROID SIGNALING KINASE 3 (BSK3) completely suppressed the response of PR but not of LRs[24]. These results indicate that additional signaling or regulatory components mediate N-dependent LR elongation.

Using natural variation and genome-wide association (GWA) mapping, we identified genetic variation in YUC8, involved in auxin biosynthesis, as determinant for the root foraging response to low N. We show that low N transcriptionally upregulates YUC8, together with its homologous genes and with TAA1, encoding a tryptophan amino transferase catalyzing the preceding step to enhance local auxin biosynthesis in roots. Genetic analysis and pharmacological approaches allowed placing local auxin production in LRs downstream of BR signaling. Our results reveal the importance of hormonal crosstalk in LRs where BRs and auxin act synergistically to stimulate cell elongation in response to low N availability.

## Results

### GWAS uncovers YUC8 as determinant for LR response to low N.
In order to identify further genetic components involved with the response of LRs to low N, we assessed LR length in a geographically and genetic diverse panel[24] of 200 A. thaliana accessions grown under high N (HN; 11.4 mM N) or low N (LN; 0.55 mM N). After transferring 7-day-old seedlings precultured on sufficient N to HN or LN for 9 days, we observed substantial phenotypic variation for average LR length among tested accessions, ranging from 0.20 to 0.80 cm at HN and from 0.43 to 1.48 cm at LN (Fig. 1a, b and Supplementary Data 1). Although LR length of all examined accessions increased when plants were grown on LN (Fig. 1b), the extent of this response (i.e., the LN-to-HN ratio of average LR length) differed substantially from 22% increase as in accession Co to 188% increase in Par-3 (Fig. 1b, c). We then performed a GWA study and detected two SNPs on chromosome 4 at positions 2724898 and 14192732, respectively, that were significantly associated (false discovery rate at $q = 0.05$) with LR response to LN (Fig. 1d). We focused on the SNP_Chr4_14192732, as the corresponding peak was supported by adjacent markers and T-DNA insertion lines were available for all genes falling within a 20-kb supporting interval. The T-variant of this lead SNP was present in 75% of the phenotyped accessions and was associated with longer LRs under LN as compared with the A-variant (Supplementary Fig. 1a), indicating that this locus might control LR growth under LN. The SNP_Chr4_14192732 was directly located in At4g28720 (Fig. 1e), which encodes the auxin biosynthesis protein YUCCA8 (YUC8). We then analyzed T-DNA insertion lines of YUC8 and another two genes (At4g28730 and At4g28740) located within the 20-kb interval centered around the identified SNP (Fig. 1e). Knockout lines of At4g28730 and At4g28740 exhibited LN-induced LR length comparable to wild-type plants, and the expression of these two genes did not respond to LN (Supplementary Fig. 1b–e), excluding an eventual role of At4g28730 and At4g28740 in regulating LR elongation induced by mild N deficiency. By contrast, loss of YUC8 expression significantly impaired the LR response to LN (Fig. 1f, h). In two independent YUC8 mutants, average LR length was similar to wild type at HN, while at LN LRs were 25% and 18% shorter in yuc8-1 and yuc8-2 plants respectively, compared to wild-type plants. Since no significant change of PR length and LR number was observed at either N condition (Fig. 1g and Supplementary Fig. 2a), the overall decrease in total root length of yuc8 mutant plants at LN was exclusively due to decreased LR length (Supplementary Fig. 2b). Together, these results indicate that YUC8 likely underlies the trait association with SNP_Chr4_14192732.

### TAA1- and YUC5/7/8-dependent auxin synthesis increase LR elongation.
The flavin-containing monooxygenase-like proteins of the YUCCA family have been shown to catalyze the rate-limiting step of auxin biosynthesis by converting indole-3-pyruvic acid (IPyA), produced by TAA1/TARs (Tryptophan Aminotransferase of Arabidopsis 1/ Tryptophan Aminotransferase Related proteins), into indole-3-acetic acid (IAA)[26–28]. Since YUC8 acts redundantly with its closest homologs[29], we assessed root architectural traits in single mutants for two additional root-expressed YUC genes (i.e., YUC 5 and 7) and in the yuc3,5,7,8,9 quintuple mutant (yucQ). The length of PRs and LRs under N deficiency was also significantly decreased in yuc5 and yuc7 mutants (Supplementary Figs. 3 and 4). In yucQ plants, low N-induced PR and LR elongation was even completely abolished (Fig. 1i–k). Aside from defective root elongation, yucQ plants also formed significantly less LRs irrespective of the N condition (Supplementary Fig. 5). Microscopic analyses revealed that loss of the LR response to LN in yucQ plants was primarily associated with attenuated cell elongation (Fig. 2a–d). To further ascertain that auxin deficiency caused the inability of yucQ roots to respond to low N, we exogenously supplied IAA to the growth medium. Consistent with the previous studies[30], PR length gradually decreased with increasing IAA supplementation in wild-type and yucQ plants (Supplementary Fig. 6a, b). However, most notably,

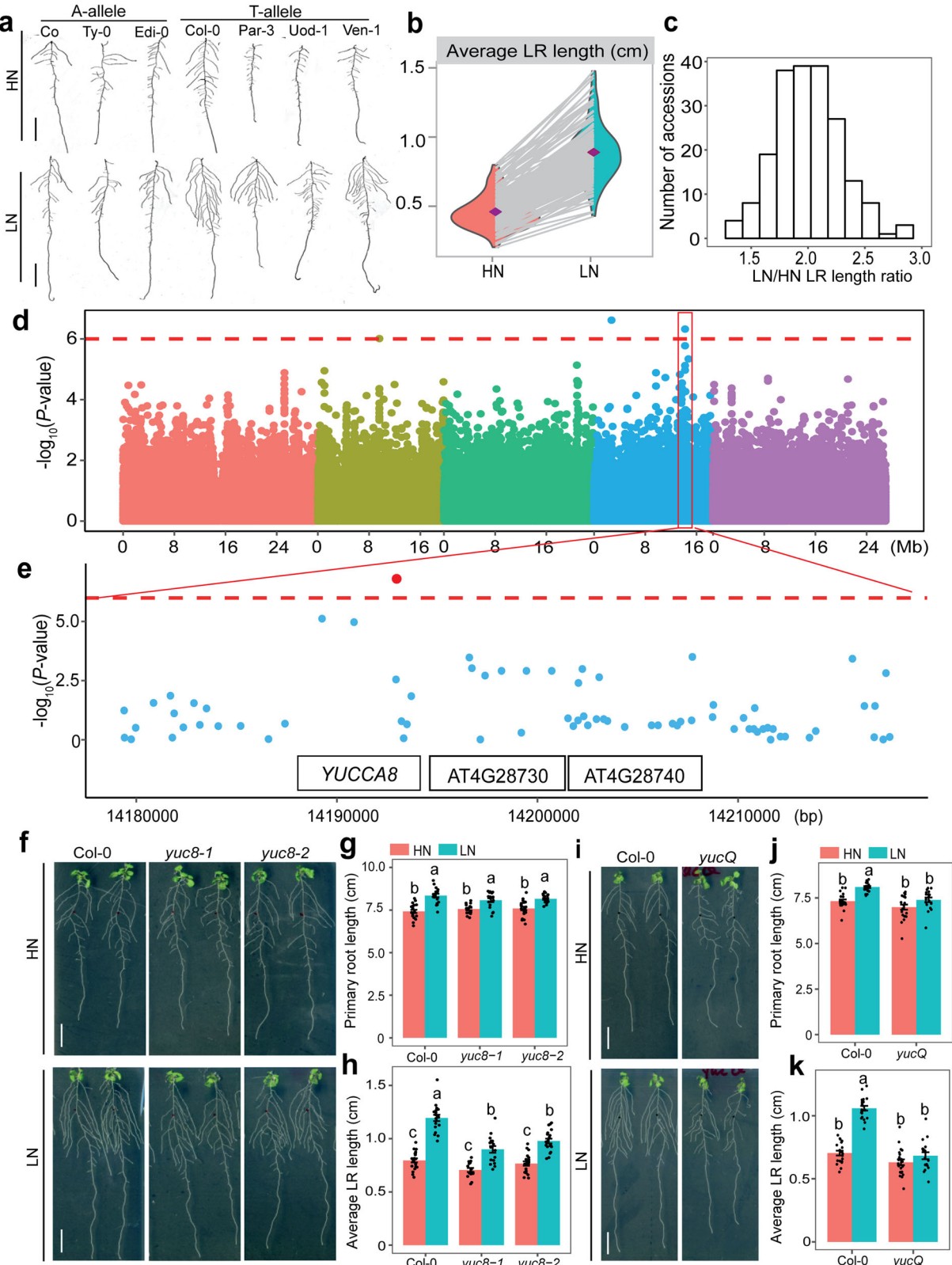

the response of PR and especially LRs of *yucQ* plants to LN was fully recovered by supplying 50 nM IAA (Supplementary Fig. 6b–c). Conversely, when YUCCA-dependent auxin biosynthesis in roots of wild-type plants was suppressed with 4-phenoxyphenylboronic acid (PPBo), a potent inhibitor of YUCCA activity[31], low N-induced elongation of both PR and LRs was strongly reduced (Supplementary Fig. 7).

As the expression of *TAA1* is upregulated by moderate N limitation in roots[21] (Supplementary Fig. 8), we then investigated if also TAA1 is required for root growth responses to mild N deficiency. Similar to *yucQ* plants, low N-induced elongation of PR and LRs were also strongly impaired in two independent *taa1* mutants (Supplementary Fig. 9). To further test the role of local auxin biosynthesis in roots for N-dependent root foraging responses, we

**Fig. 1 Natural variation of the LR response to low N and GWA mapping of YUC8. a** Representative A- and T-allele accessions of *A. thaliana* that show weak (Co, Ty-0, Edi-0), intermediate (Col-0), and strong (Par-3, Uod-1, Ven-1) LR elongation response to low N availability. HN, high N (11.4 mM N); LN, low N (0.55 mM N). **b** Reaction norms and phenotypic variation of average LR length of 200 natural accessions of *A. thaliana* under different N supplies. Purple diamonds represent the means of lateral root lengths for 200 accessions under each N treatment. **c** Frequency distribution of LR response to N availability (i.e., the ratio between LN and HN) for 200 natural accessions. **d** Manhattan plot for SNP associations with LR response to low N performed with vGWAS package. Negative $\log_{10}$-transformed *P* values from a genome-wide scan were plotted against positions on each of the five chromosomes of *A. thaliana*. Chromosomes are depicted in different colors (I to V, from left to right). The red dashed line corresponds to the Benjamini and Hochberg false-discovery rate level of *q* < 0.05 adjusted for multiple testing. **e** The 20-kb-long genomic region concentered on the lead GWA peak for LR response to low N, and genes located within this region. **f–h** Appearance of plants (**f**), primary root length (**g**), and average LR length (**h**) of wild-type (Col-0) and two *yuc8* mutants. Bars represent means ± SEM. Number of individual roots analyzed in HN/LN: *n* = 20/19 (Col-0), 15/17 (*yuc8-1*), 20/20 (*yuc8-2*). **i–k** Appearance of plants (**i**), primary root length (**j**), and average LR length (**k**) of wild-type (Col-0) and *yucQ* mutant after 9 days on HN or LN. Bars represent means ± SEM. Number of individual roots analyzed in HN/LN: *n* = 20/21 (Col-0) and 22/17 (*yucQ*). Different letters in (**g**, **h**) and (**j**, **k**) indicate significant differences at *P* < 0.05 according to one-way ANOVA and post hoc Tukey test. Scale bars, 1 cm.

supplied the polar auxin transport inhibitor *N*-1-naphthylphthalamic acid (NPA) to the shoots in a split-agar setup (Supplementary Fig. 10). Our results showed that LR response to low N was not significantly inhibited when shoot-to-root auxin translocation was blocked. Collectively, these results indicate that TAA1- and YUC5/7/8-mediated local auxin production in roots modulates root elongation under mild N deficiency.

Previously, it has been shown that the transcription factor *AGL21* is required for sustaining LR elongation in N-free media, and that auxin accumulation in LRs and the expression of multiple *YUC* genes can be altered by *AGL21* mutation or overexpression under non-stressed conditions[20]. We then investigated whether *AGL21* and its close homologous gene *ANR1* also control systemic stimulation of LR elongation by mild N deficiency. We found that the *agl21 anr1* double mutant exhibits comparable root foraging responses to mild N deficiency as wild-type plants (Supplementary Fig. 11). These results suggest that distinct mechanisms modulate foraging *versus* survival responses in roots. In support of this notion, roots of *yuc8* or *yucQ* mutants responded to N starvation similarly to wild-type plants (Supplementary Figs. 12 and 13), indicating that survival responses to low N are likely independent of YUCCA-dependent local auxin biosynthesis in roots.

**Low N enhances YUC3/5/7/8 to increase auxin in LR tips.** We next investigated whether external N availability regulates the expression of root-expressed *YUC* genes. Similar to *TAA1*, mRNA levels of *YUC8*, *YUC3*, *YUC5* and *YUC7* were also significantly upregulated by low N (Fig. 2e–h). N-dependent regulation of *YUC8* was confirmed by assessing *YUC8* promoter activity in the meristems of PR and LRs (Fig. 2i and Supplementary Fig. 14a, b). Whereas previous studies have shown that low N availability increases auxin levels in roots[32–34], our results indicated that this relies on a YUCCA-dependent increase in local auxin biosynthesis. To further test this assumption, we monitored auxin accumulation with the ratiometric auxin sensor R2D2[35]. We found that DII-n3xVenus/mDI-ntdTomato ratio decreased in both PR and LR tips of low N-grown plants, which is indicative of higher auxin accumulation (Fig. 2j, k, and Supplementary Fig. 14c, d). Inhibition of YUCCAs by the supply of PPBo to roots substantially reverted low N-induced auxin accumulation (Fig. 2j, k and Supplementary Fig. 14c, d), thus corroborating the critical role of YUCCAs in enhancing local auxin biosynthesis and stimulating root elongation under mild N deficiency.

**Allelic coding variants of YUC8 determine LR foraging.** Our GWA mapping and genetic analyses indicated that allelic variation in *YUC8* is linked to phenotypic variation of LR growth. Expression levels of *YUC8* at HN and LN or expression changes

in representative natural accessions with contrasting LR responses to LN were neither significantly correlated with average LR length nor with the LR response to LN (Supplementary Fig. 15). These results suggested that YUC8-dependent natural variation under LN is likely not due to variations at the transcript level. We then searched for SNPs within *YUC8*'s coding sequence from 139 re-sequenced lines from our original panel and detected 17 SNPs (MAF > 5%), all of which result in synonymous substitutions, except for two SNPs (T41C and A42T) that together result in a non-synonymous substitution from leucine (L) to serine (S) at position 14 (Supplementary Data 2). This non-synonymous substitution is 14 amino acids away from the FAD-binding motif, which is critical for YUC8 activity[36,37]. A generalized linear model association analysis of average LR length with these polymorphic sites showed that 6 of them were significantly associated with average LR length only at LN but not at HN (Fig. 3a). These 6 SNPs allowed us to group accessions into two major haplotypes (Supplementary Data 3), with YUC8-hap A (TAGCAA) associated with longer and YUC8-hap B (CTATGG) with shorter LRs at LN (Fig. 3b). Consequently, total LR length and total root length were on average longer in YUC8-hap A than YUC8-hap B accessions (Supplementary Fig. 16).

To test the causality of the two identified YUC8 variants, we placed the coding sequence of *YUC8* from Col-0 (YUC8-hap A) or Co (YUC8-hap B) downstream of the $YUC8_{Col-0}$ promoter and expressed the constructs in the *yucQ* mutant (Fig. 3c). We initially observed that the short PR length and decreased growth rate of *yucQ* plants were rescued more efficiently by expressing the YUC8-hap A variant than YUC8-hap B (Supplementary Fig. 17). We then tested whether allelic variation in YUC8 is indeed relevant for root growth in the context of N deficiency. Consistent with our haplotype analysis (Fig. 3b), T2 *yucQ* plants expressing YUC8-hap A displayed longer PR and LRs than those expressing YUC8-hap B (Fig. 3d–g). To rule out possible effects of differential *YUC8* expression due to random genomic integration of the expression cassette, we further assessed three independent T3 homozygous lines for each variant showing comparable *YUC8* expression levels (Supplementary Fig. 18a). Also in these lines complementation of PR, LR, and total root length at LN was more efficient with YUC8-hap A than with YUC8-hap B (Fig. 4a–c and Supplementary Fig. 18b). Consequently, root foraging responses induced by mild N deficiency were significantly stronger in lines expressing the YUC8-hap A variant than in those expressing the YUC8-hap B (Supplementary Fig. 18c–e). Microscopic analyses suggested that the stronger LR foraging response conferred by YUC8-hap A was primarily due to increased cell elongation (Fig. 4d, e), while meristem size made a minor contribution (Fig. 4f and Supplementary Fig. 19). We then tested if the differential auxin biosynthesis drives the divergent root foraging responses between YUC8-hap A and -hap B accessions by inhibiting the activities of YUCCAs in roots with PPBo. Whereas

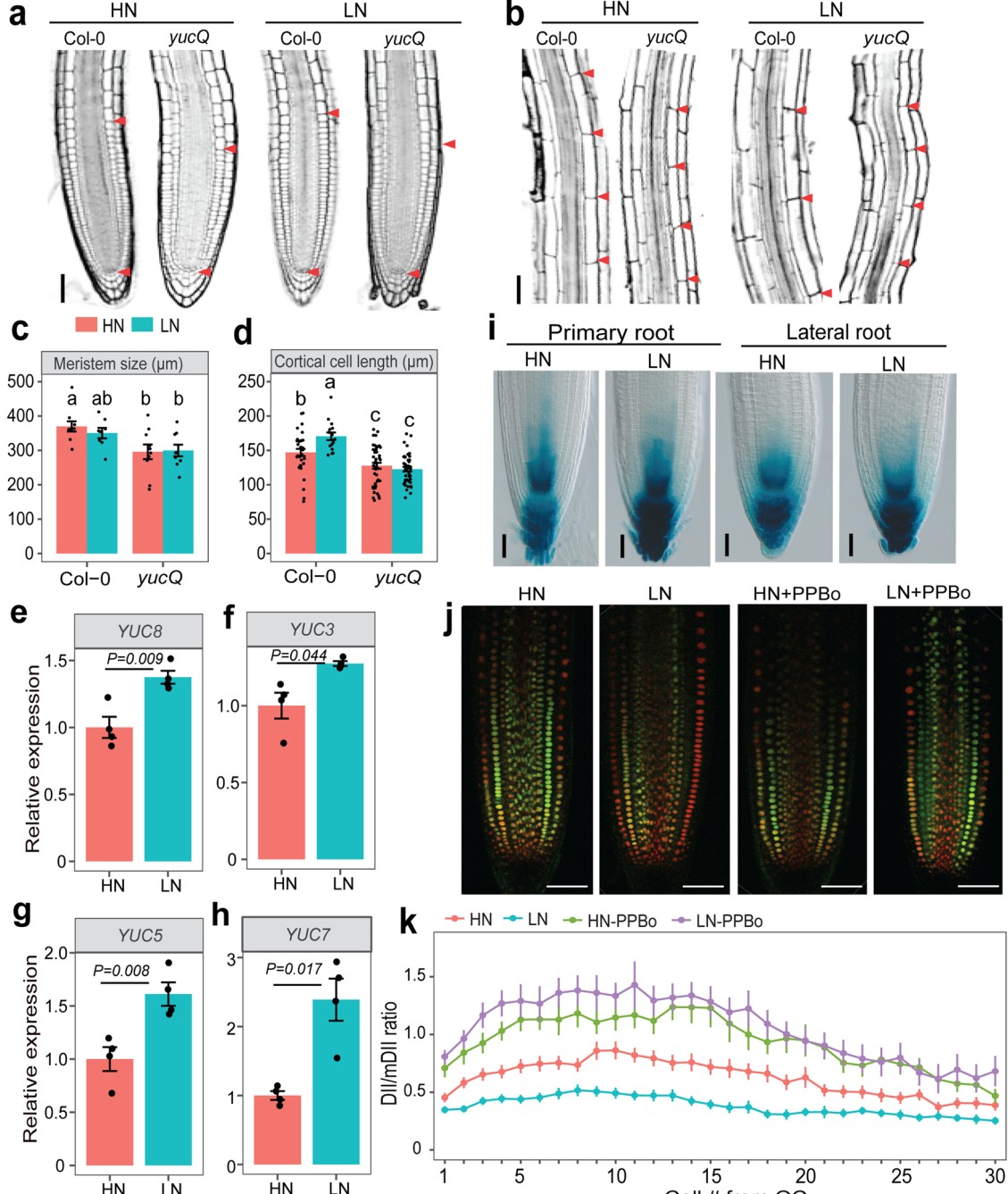

**Fig. 2 YUCCA-dependent auxin biosynthesis is required to stimulate LR elongation under low N. a–b** Representative confocal images of root meristems (**a**) and mature cells (**b**) of Col-0 and *yucQ* LRs grown under high N (HN, 11.4 mM N) or low N (LN, 0.55 mM N). Red arrowheads indicate the position of the quiescent center (QC) and the boundaries between the meristematic and elongation zones (**a**) or between two consecutive mature cortical cells (**b**). Scale bars, 50 μm. **c–d** Length of the meristem (**c**) and cortical cells (**d**) of LRs from Col-0 and *yucQ* plants grown under HN or LN. Bars represent means ± SEM. Number of individual roots or cells analyzed in HN/LN: *n* = 10/8 (Col-0) and 10/9 (*yucQ*) in (**c**); 34/16 (Col-0) and 45/43 (*yucQ*) in (**d**). Different letters indicate significant differences at *P* < 0.05 according to one-way ANOVA and post hoc Tukey test. **e–h** Transcript levels of *YUC8* (**e**), *YUC3* (**f**), *YUC5* (**g**), *and YUC7* (**h**) in response to HN and LN. Root samples for *q*PCR analysis were taken 9 days after transfer. Expression levels were assessed in whole roots by *q*PCR analysis and normalized to *ACT2* and *UBQ10*. Bars represent means ± SEM (*n* = 4 independent biological replicates). *P* values relate to differences between two N conditions according to Welch's *t*-test. **i** *proYUC8*-dependent GUS activity in the tips of primary root (left panel) and LR (right panel) at 9 days after transfer to HN or LN. Scale bars, 100 μm. **j** Representative images of mDII-ntdTomato and DII-n3xVenus in tips of mature LRs grown HN or LN and supplemented with 5 μM YUCCA activity inhibitor 4-phenoxyphenyl boronic acid (PPBo). **k** DII-n3xVenus/mDII-ntdTomato intensity ratio in epidermal cells of mature LRs. The experiments in (**a**, **b**) and (**i**, **j**) were repeated twice with similar results. Dots represent means ± SEM (*n* = 30, 25, 15, and 15 roots for HN, LN, HN-PPBo and LN-PPBo, respectively). Scale bars, 100 μm.

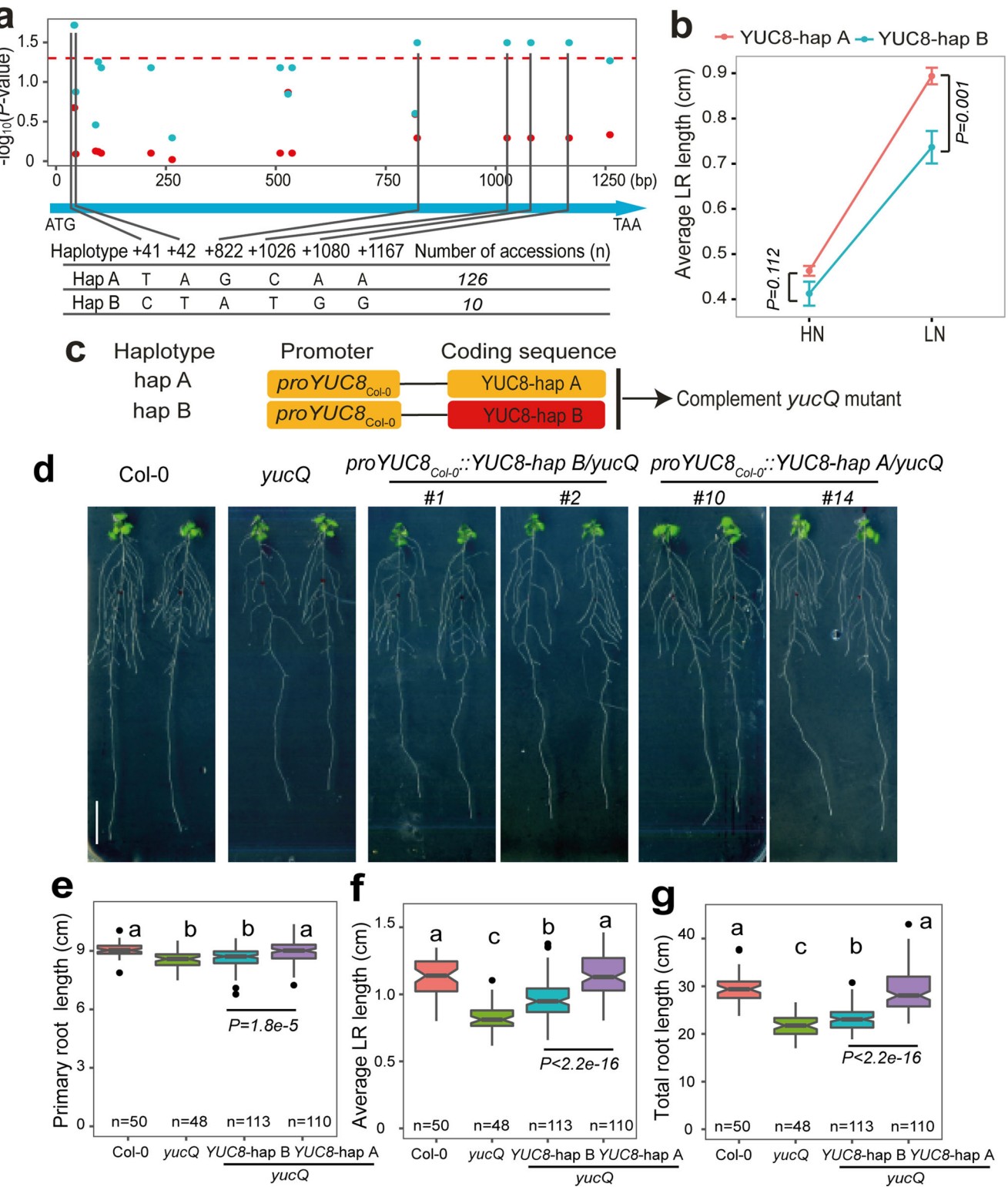

the selected accessions exhibited the expected differential root responsiveness to low N (i.e. LN-to-HN ratio) under mock conditions, exogenous supply of PPBo to roots completely eliminated the strong foraging response of YUC8-hap A accessions (Supplementary Fig. 20). Altogether, these data corroborated that natural variation in the coding sequence of *YUC8* and YUCCA-dependent root auxin accumulation determines the extent of the root foraging response to mild N deficiency.

**Auxin tunes LR foraging downstream of BR signaling**. Our previous work showed that BR biosynthesis and signaling are involved in regulating root elongation under low N[24,25]. We then explored a potential interdependence and hierarchy in auxin- and BR-dependent coordination of LR elongation in response to LN. Therefore, we generated a *bsk3 yuc8* double mutant, which showed significantly shorter LRs than the wild type under LN but no additive effect compared to the single mutants *bsk3* and *yuc8*

**Fig. 3 Allelic variants of YUC8 cause variation in LR length at low N. a** Association of 17 polymorphic sites (MAF > 0.05) in the coding region of single-exon gene *YUC8* in 139 re-sequenced accessions with average LR length under high N (HN, 11.4 mM N; red) or low N (LN, 0.55 mM N; cyan). The *x*-axis shows the nucleotide position of each variant. The y-axis shows the $-\log_{10}$ (*P*-value) for the association test using a generalized linear model (GLM), with a significance level at $\alpha = 0.05$ indicated with a dashed red line. The six polymorphisms selected for further analysis were projected onto a schematic representation of a *YUC8* gene structure represented by a light blue arrow. **b** Average LR length of natural accessions representing two major *YUC8* haplotypes (*n* = 126 and 10 accessions for haplotype A and haplotype B, respectively). Dots represent means ± SEM and *P* values relate to differences between two haplogroups under respective N conditions according to Welch's *t*-test. **c** Schematic of *YUC8* constructs to complement the *yucQ* mutant. **d–g** Root phenotype of transgenic allelic complementation lines at low N. Appearance of plants (**d**), PR length (**e**), average LR length (**f**), and total root length (**g**) at LN of wild-type (Col-0), *yucQ* and independent transgenic plants expressing sequences coding for either *YUC8*-haplotype A or *YUC8*-haplotype B under control of the $YUC8_{Col-O}$ promoter. Six independent T2 lines for each construct were assessed. Two representative lines are shown for each construct. Root system architecture was assessed after 9 days. Horizontal lines show medians; box limits indicate the 25th and 75th percentiles; whiskers extend to 1.5 times the interquartile range from the 25th and 75th percentiles. Numbers below each box indicate the number of plants assessed for each genotype under the respective N condition. Different letters in (**e–g**) indicate significant differences at *P* < 0.01 according to one-way ANOVA and post hoc Tukey test. *P* values relate to differences between two complementing groups according to Welch's *t*-test. Scale bar, 1 cm.

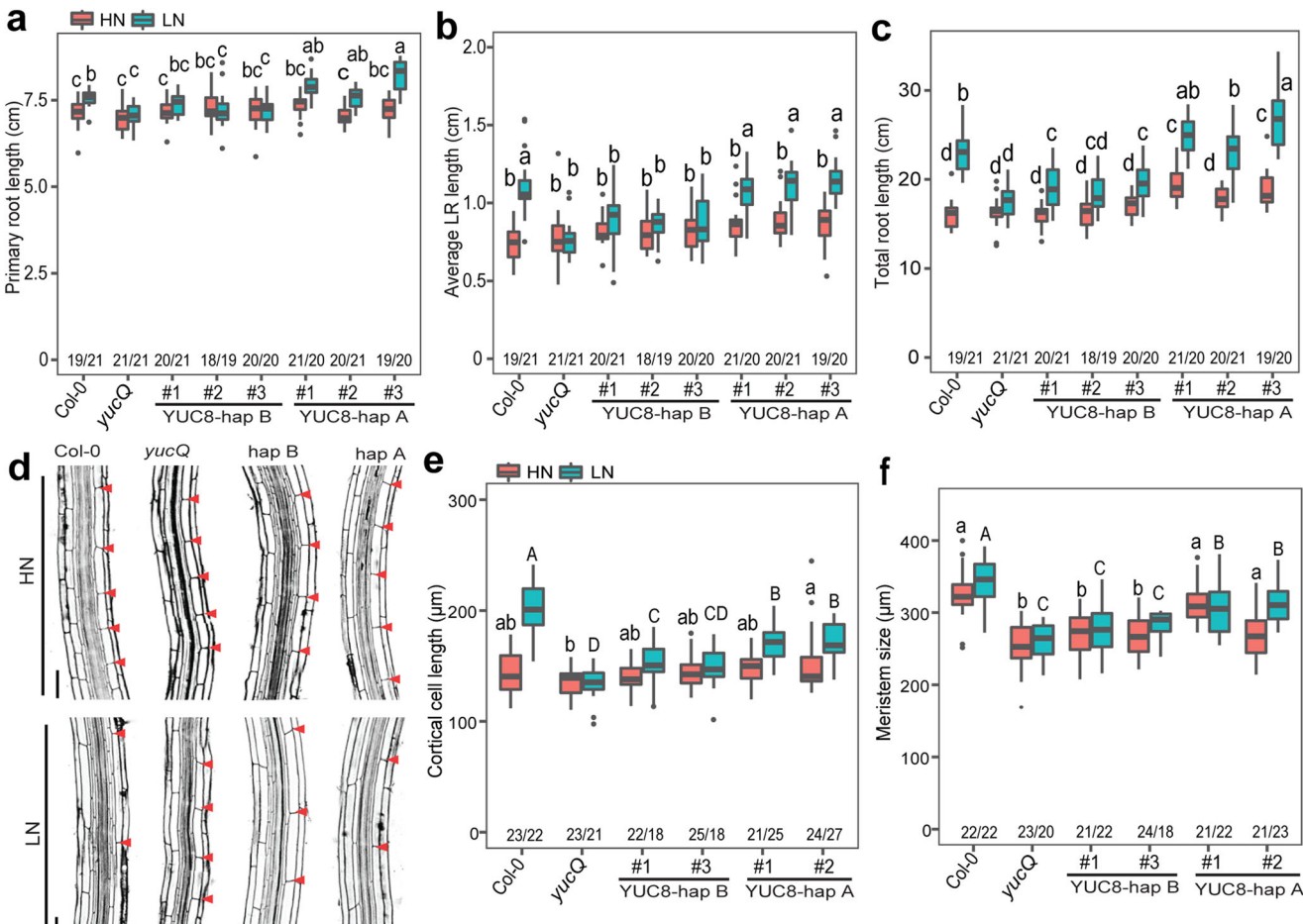

**Fig. 4 Allelic variants of YUC8 determine the extent of root foraging for N. a–c** Primary root length (**a**), average LR length (**b**), and total root length (**c**) of wild-type (Col-0), *yucQ* and three independent transgenic lines expressing sequences coding for either the *YUC8*-hap A or *YUC8*-hap B under control of the $YUC8_{Col-O}$ promoter. **d** Representative confocal images of cortical cells of mature LRs of wild-type (Col-0), *yucQ* and transgenic lines complemented with either *YUC8* variants under control of the $YUC8_{Col-O}$ promoter grown under high N (HN, 11.4 mM N) or low N (LN, 0.55 mM N). Red arrowheads indicate the boundary between two consecutive cortical cells. One representative line was shown for each construct. Scale bars, 50 μm. **e–f** Length of cortical cells (**e**) and meristems (**f**) of LRs of wild-type (Col-0), *yucQ* and complemented *yucQ* lines grown under HN or LN for 9 days. The experiment was repeated twice with similar results. Horizontal lines show medians; box limits indicate the 25th and 75th percentiles; whiskers extend to 1.5 times the interquartile range from the 25th and 75th percentiles. Numbers below each box indicate the number of plants assessed for each genotype under respective N condition. Different lowercase letters at HN and uppercase letters at LN indicate significant differences at *P* < 0.05 according to one-way ANOVA and post hoc Tukey test.

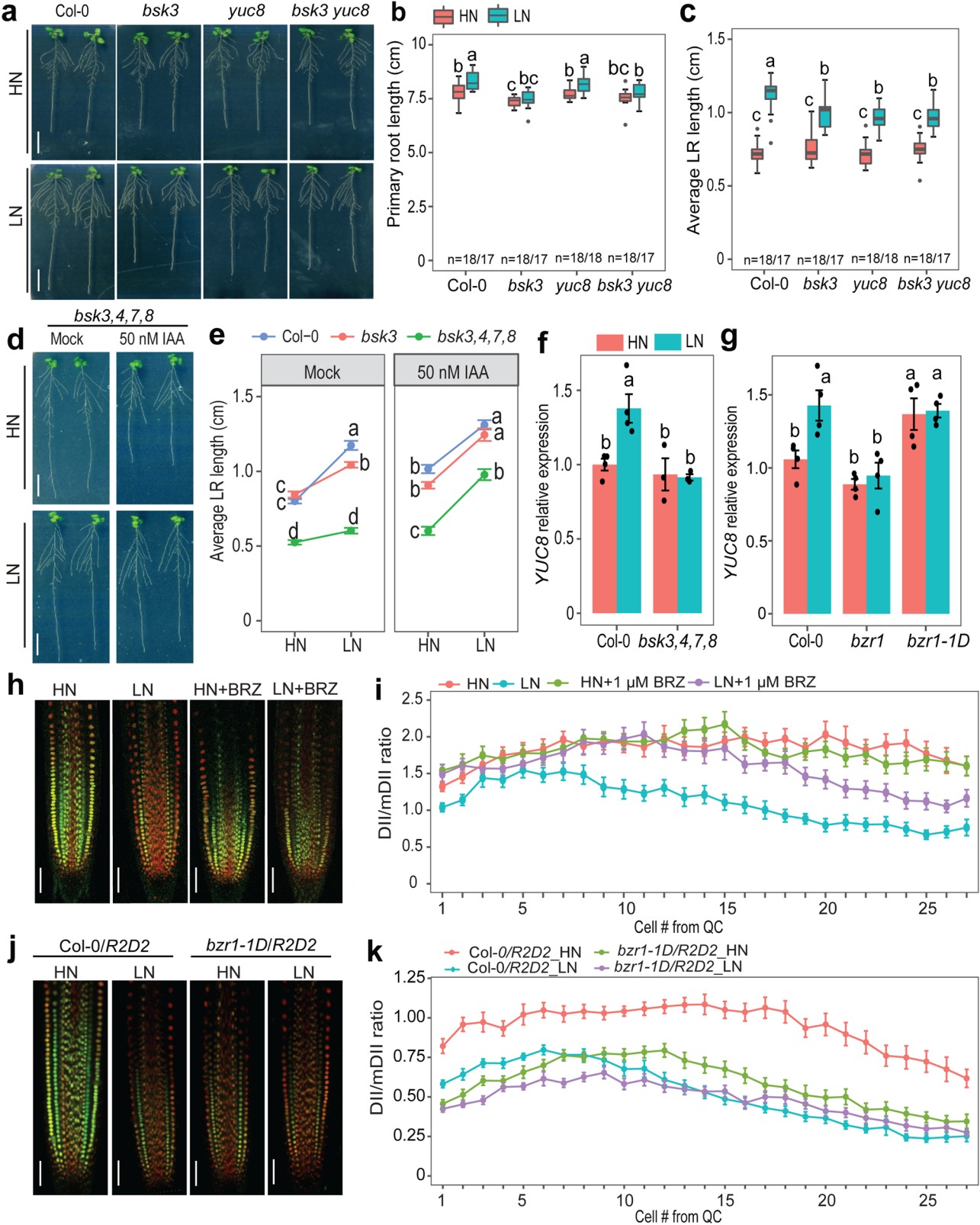

(Fig. 5a–c). This result suggested that BSK3 and YUC8 act in the same signaling route to modulate LR elongation at LN. Consistent with our previous observation that BR sensitivity increases in N-deficient roots[24], exogenous application of brassinolide (the most bioactive BR) gradually suppressed the LR response to LN of wild-type plants (Supplementary Fig. 21). However, in the *yucQ* mutant, the response of LRs to LN was largely insensitive to exogenous BR supplies. In contrast, the LR foraging response to LN of the BR signaling mutants *bsk3* and *bsk3,4,7,8* as well as of the BR biosynthesis mutant *dwf4-44* was restored under exogenous application of IAA (Fig. 5d, e and Supplementary Fig. 22). These results reveal a dependency of local auxin biosynthesis in LRs on BR function and place local auxin biosynthesis downstream of BR signaling.

**Fig. 5 Auxin biosynthesis acts epistatic to and downstream of BR signaling to regulate LR response to low N. a–c** Appearance of plants (**a**), primary root length (**b**) and average lateral root length (**c**) of wild-type (Col-0), *bsk3*, *yuc8* and *bsk3 yuc8* plants grown under high N (HN, 11.4 mM N) or low N (LN, 0.55 mM N). Horizontal lines show medians; box limits indicate the 25th and 75th percentiles; whiskers extend to 1.5 times the interquartile range from the 25th and 75th percentiles. Numbers below each box indicates the number of plants assessed for each genotype under the respective N condition. **d** Appearance of *bsk3,4,7,8* mutant plants grown at HN or LN in the presence or absence of 50 nM IAA. **e** The LR response of *bsk3* and *bsk3,4,7,8* plants to low N is rescued in presence of exogenous IAA. Dots represent means ± SEM. Number of individual roots analyzed in HN/LN: $n = 19/22$ (mock) and 17/17 (50 nM IAA) for Col-0; 15/15 (mock) and 17/17 (50 nM IAA) for *bsk3*; 17/16 (mock) and 18/18 (50 nM IAA) for *bsk3,4,7,8*. Average LR length was assessed 9 days after transfer. **f–g** Transcript levels of *YUC8* in *bsk3,4,7,8* (**f**) and *BZR1* loss- (*bzr1*) or gain-of-function (*bzr1-1D*) mutants (**g**). Expression levels were assessed in roots by qPCR and normalized to *ACT2* and *UBQ10*. Bars represent means ± SEM ($n = 4$ for Col-0, *bzr1*, *bzr1-1D*, and three independent biological replicates for *bsk3,4,7,8* at both N conditions). **h–i** Representative images (**h**) and ratio of mDII-ntdTomato and DII-n3xVenus fluorescence signals (**i**) in mature LR tips of wild-type plants grown for 7 days on HN or LN in the presence or absence of 1 μM brassinazole, a BR biosynthesis inhibitor. **j–k** Representative images (**j**) and ratio of mDII-ntdTomato and DII-n3xVenus fluorescence signals (**k**) in mature LR tips of *Col-0/R2D2* and *bzr1-1D/R2D2*. In (**h–j**), Scale bars, 100 μm. In (**h–k**), DII-n3xVenus and mDII-ntdTomato fluorescence was quantified in epidermal cells of mature LRs. Dots represent means ± SEM ($n = 20$ roots). Different letters in (**b**, **c**, **e–g**) indicate significant differences at $P < 0.05$ according to one-way ANOVA and post hoc Tukey test.

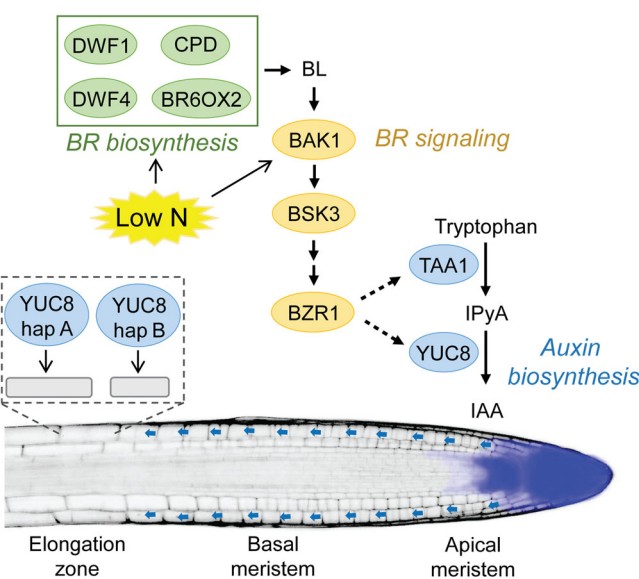

**Fig. 6 Model for low N-induced local auxin biosynthesis downstream of BR signaling to stimulate LR elongation.** Low external N availability that results in mild N deficiency induces the expression of the BR co-receptor *BAK1* (Jia et al.[24]) and several genes involved in BR biosynthesis (Jia et al.[25]). Downstream of BR signaling, an auxin biosynthesis module composed of TAA1 and YUC8 together with its homologs YUC5 and YUC7 is induced to generate more IAA in the apical meristem of LRs (blue area in LR). Upon transport to the elongation zone (blue arrows), locally generated IAA enhances cell expansion. Allelic coding variants of YUC8 in natural accessions of *A. thaliana* determine the extent of the root foraging response to low N by differentially modulating cell elongation (schematic representation within dashed box).

To further explore how BR signaling regulates auxin biosynthesis, we analyzed the N-dependent expression of *YUC5*, *YUC7*, and *YUC8* in the *bsk3,4,7,8*, *bzr1*, and *bzr1-1D* mutants. Whereas the expression of these *YUC* genes was not significantly altered at HN, they were not anymore upregulated by LN in *bsk3,4,7,8* and *bzr1* roots (Fig. 5f, g and Supplementary Fig. 23). Likewise, LN-induced upregulation of *TAA1* was also lost in the *bzr1* mutant (Supplementary Fig. 8). Interestingly, in *bzr1-1D* mutant plants, which carry a stabilized variant of the BZR1 transcription factor[38], *TAA1*, *YUC7* and *YUC8* were upregulated irrespective of the N regime (Fig. 5g and Supplementary Figs. 8 and 23d). Next, we assessed if BRs stimulate auxin accumulation in LR meristems by assessing auxin levels with the *R2D2* reporter

after the supply of the potent BR biosynthesis inhibitor brassinazole[39] (BRZ), or in the *bzr1-1D* mutant with constitutively active BR signaling[38]. Supply of 1 μM BRZ, a concentration that can largely inhibit low N-induced LR elongation[24,25], increased the DII/mDII ratio under low N (Fig. 5h, i), indicating less auxin accumulation. In contrast, the DII/mDII ratio strongly decreased in LRs of *bzr1-1D* irrespective of available N, suggesting that constitutive activation of BR signaling can increase auxin levels in LRs (Fig. 5j, k). Taken together, these data suggest that LN-induced LR elongation relies on BR signaling-dependent upregulation of *TAA1* and *YUC5/7/8* expression to increase local auxin biosynthesis.

## Discussion

Root developmental plasticity is crucial for plant fitness and nutrient capture. When encountering low external N availability that induces mild N deficiency, plants from several species enlarge their root systems by stimulating the elongation of LRs[18,21–23]. Here we show that coding variation in the *YUC8* gene determines the extent of LR elongation under mild N deficiency and that *TAA1*- and *YUC5/7/8*-dependent local auxin biosynthesis acts downstream of BR signaling to regulate this response (Fig. 6). Our findings not only provide insights into how auxin homeostasis itself is subject to natural variation, but uncovered a previously unknown crosstalk between BRs and auxin that coordinates morphological root responses to N deficiency.

While previous studies have shown that auxin levels increase in roots of N-deficient plants[32–34], the source of this auxin and its contribution to low N-induced root elongation still remained unresolved. Our results show that mild N deficiency stimulates local auxin accumulation in the root apical meristem by upregulating *TAA1* and a set of *YUCCA* genes (Fig. 6). We also raised further evidence that the signaling pathways involved with root foraging responses induced by moderate N deficiency are distinct from those required to alter root growth under N starvation, i.e. in absence of N (Fig. 1f–k and Supplementary Figs. 11–13). With the help of GWA mapping, we found that natural variants of *YUC8* significantly contribute to LR elongation under mild N deficiency. YUC8 belongs to the family of flavin-containing monooxygenases (FMO), which use NADPH as electron donor and FAD as cofactor to convert IPyA to IAA[37]. Previously, it has been shown that a subset of YUCs, including YUC8, possesses an N-terminal signal anchor and colocalizes with the endoplasmic reticulum (ER)[40]. Our genetic analyses showed that expression of the YUC8-hap A coding variant conferred an overall improved root growth compared to YUC8-hap B (Figs. 3, 4 and Supplementary Figs. 17–19). In a small set of accessions, we detected two mutations ($T_{41}A_{42} \rightarrow C_{41}T_{42}$) in the coding region of YUC8 which

confer a non-synonymous substitution of leucine (L) to serine (S) at position 14. Unfortunately, a quantitative assessment of the in vitro catalytic properties of the two YUC8 proteoforms has remained technically challenging, as the production of sufficient quantities of soluble proteins has failed so far. Such difficulty is common for proteins associated with the ER membrane[37,41,42]. While the L to S substitution found here lies outside the critical FAD domain, it could potentially affect YUC8 activity by changing hydrophilicity or providing a putative phosphorylation site. However, so far post-translational regulation of auxin biosynthesis by phosphorylation has only been reported for TAA1[43] but not for YUCs.

As *A. thaliana* colonizes a wide range of different environments, part of the genetic variation and the resulting phenotypic variation could be associated with adaptive responses to local environments[44,45]. For example, it has been recently shown that natural allelic variants of the auxin transport regulator *EXO70A3* are associated with rainfall patterns and determine adaptation to drought conditions[46]. We found that the top GWAS SNP from our study is most significantly associated with temperature seasonality and that the distribution of YUC8-hap A and -hap B variants is highly associated with temperature variability (Supplementary Fig. 24), suggesting that YUC8 allelic variants may play an adaptive role under temperature fluctuations. This possibility is supported by previous findings that YUC8-dependent auxin biosynthesis is necessary to stimulate hypocotyl and petiole elongation in response to increased air temperatures[47,48]. However, to what extent this putative evolutionary adaptation is related to the identified SNPs in YUC8 remains to be investigated.

Our results further demonstrate that BR levels and signaling regulate local, TAA1- and YUC5/7/8-dependent auxin production especially in LRs. Microscopic analysis indicated that mild N deficiency stimulates cell elongation in LRs, a response that can be strongly inhibited by genetically perturbing auxin synthesis in roots (Fig. 2a–d). This response resembles the effect of BR signaling that we uncovered previously[24] and suggested that the coordination of root foraging response to low N relies on a genetic crosstalk between BRs and auxin. These two plant hormones regulate cell expansion in cooperative or even antagonistic ways, depending on the tissue and developmental context[49–52]. In particular, BR has been shown to antagonize auxin signaling in orchestrating stem cell dynamics and cell expansion in the PRs of non-stressed plants[49]. Surprisingly, in the context of low N availability, these two plant hormones did not act antagonistically on root cell elongation. Instead, our study uncovered a previously unknown interaction between BRs and auxin in roots that resembles their synergistic interplay to induce hypocotyl elongation in response to elevated temperatures[50–52]. Genetic analysis of the *bsk3 yuc8* double mutant showed a non-additive effect on LR length compared to the single mutants *bsk3* and *yuc8-1* (Fig. 5a–c), indicating auxin and BR signaling act in the same pathway to regulate LR elongation under low N. Whereas the exogenous supply of BR could not induce LR elongation in the *yucQ* mutant under low N (Supplementary Fig. 21), exogenous supply of auxin to mutants perturbed in BR signaling or biosynthesis was able to restore their LR response to low N (Fig. 5d, e and Supplementary Fig. 22). These results collectively indicate that BR signaling regulates auxin biosynthesis at low N to promote LR elongation. Indeed, the expression levels of *TAA1* and *YUC5/7/8* were significantly decreased at low N in BR signaling defective mutants (Fig. 5f, g and Supplementary Figs. 8 and 23). Notably, when BR signaling was perturbed or enhanced, low N-induced auxin accumulation in the root apex was significantly compromised or increased, respectively (Fig. 5h–k). Together, these results established the dependency of BR functions on auxin biosynthesis. Although our results placed local auxin biosynthesis downstream

of BR signaling (Fig. 5 and Supplementary Figs. 21–23), this signaling cascade is likely not linear and may entail a positive feed-back loop, as auxin has been shown to stimulate BR biosynthesis in roots by inducing *DWF4* expression[53]. Furthermore, our data support the view that the increased auxin produced in the apical meristem of N-deficient roots does not only counterbalance the growth-suppressive effect of elevated BR levels in the root apical meristem but also directly stimulates cell expansion in the elongation zone. Future studies may address how this local, N-responsive BR-auxin module is regulated by systemic N-demand signals and why N deficiency-induced elongation of LRs is more sensitive to auxin than the PR. Interestingly, LR elongation is stimulated in *cepr1* and *cepr1/2* mutants[54], suggesting that systemic N signaling via the CEP-CEPRs-CEPDs cascade could be involved in the regulation of this hormonal module uncovered in the present study. In the future, it will be interesting to examine whether the BR-auxin module also plays a role in root elongation under other abiotic stresses such as phosphorus deficiency or water deficit. Under any of these constraints, employing CRISPR-mediated gene editing to turn "weak" YUC8 variants into "strong" variants could provide an opportunity to increase root elongation and subsequent water and nutrient acquisition in crops.

## Methods

**Plant materials and growth conditions**. The *Arabidopsis thaliana* accession Col-0 and Col-3 were used as wild-types in this study. The T-DNA insertion lines *yuc8-1* (SALK_096110C, N655757), *yuc8-2* (SM_3.23299, N110939), *yuc5-1* (SAIL_116_C01, N860386), *yuc5-2* (SALK_088618C, N672844), *yuc7-1* (SALK_059832C, N659416), *yuc7-2* (SALK_034074C, N680792), *dwf4-44* (SAIL_882_F07, N839744), *ckrc1-1* (N66987), *wei8-1* (N16407), *bzr1* (SALK_208661C, N2104186) and *bzr1-1D* (N65987), SALK_077059C (N668516) and SAIL_1286_E04C (N867481), and the reporter line *R2D2* (N2105637) were purchased from Nottingham Arabidopsis Stock Center (NASC, Nottingham, United Kingdom). The *bsk3*, *bsk3,4,7,8*, *agl21 anr1*, and *yucQ* in the Col-0 background and *proYUC8-GUS* lines have been described in previous studies[24,55–57]. The *bsk3 yuc8* double mutant was generated by crossing the *bsk3* and *yuc8-1* and homozygous F3 plants were selected. Homozygotes and gene transcript levels of all lines used in the current study were confirmed by PCR and qRT-PCR using primers listed in Supplementary Data 4. The mutant lines used in the present study were described in Supplementary Data 5 and the expression levels of disrupted genes were shown in Supplementary Fig. 25.

Seeds were surface-sterilized by incubation in 70% (v/v) ethanol and 0.05% (v/v) Triton X-100 for 15 min. Seeds were sown on modified half-strength MS medium (750 μM $MgSO_4·7H_2O$, 625 μM $KH_2PO_4$, 1500 μM $CaCl_2·2H_2O$, 0.055 μM $CoCl_2·6H_2O$, 0.053 μM $CuCl_2·2H_2O$, 50 μM $H_3BO_3$, 2.5 μM KI, 50 μM $MnCl_2. 4H_2O$, 0.52 μM $Na_2MoO_4·2H_2O$, 15 μM $ZnCl_2$, 75 μM Fe-EDTA) supplemented with 11.4 mM N (1 mM $NH_4NO_3$ + 9.4 mM $KNO_3$), 0.5% (w/v) sucrose, 1% (w/v) Difco agar (Becton Dickinson) and 2.5 mM MES (pH 5.6) and then kept in the darkness at 4 °C for two days to synchronize germination. After stratification, agar plates containing seeds were placed vertically in a growth cabinet (Percival Scientific) under a 19 °C/22 °C and 14 h/10 h night/day regime with light intensity adjusted to 120 μmol photons $m^{-2}s^{-1}$. Seven-day-old seedlings of similar size were transferred to fresh plates with identical sucrose, agar, and nutrient composition as described above but supplied with either 11.4 mM (HN) or 0.55 mM (LN) N. The position of the primary root tip at the time when the seedlings were transferred from pre-culture to the N treatments was marked by black dots or small dashes. If not indicated otherwise, plants were cultivated for 9 days on these conditions. Treatments with IAA, 4-phenoxyphenyl boronic acid (PPBo), 24-epibrassinolide (BL) or brassinazole (BRZ) were performed by transferring seven-day-old seedlings to ½ MS medium supplemented with indicated concentrations of IAA (I0901, Duchefa biochemie), PPBo (CAS Number 51067-38-0, Sigma), 24-epibrassinolide (CAS Number 78821-43-9, Sigma) or brassinazole (CAS Number 280129-83-1, Sigma) dissolved in pure ethanol or DMSO. Agar plates supplied with an identical concentration of solvent served as mock treatments.

**Root phenotyping, genetic mapping, and haplotype analysis**. For collecting LR lengths, we grew 200 accessions (Supplementary Data 1) on HN and LN agar plates (4 individual plants per plate) and repeated the screening three times so that a total of 12 plants per accession were finally analyzed per accession on either N condition. Roots were scanned using an Epson Expression 10000XL scanner (Seiko Epson) with a resolution of 300 dots per inch after they were clearly separated from one another on the agar plate. Total LR length was quantified with WinRhizo Pro version 2009c (Regent Instruments) and LR number was manually counted. The average LR length was calculated by dividing total LR length by LR number. Average values calculated from 12 plants per line for 200 accessions were used as phenotypic

response in GWAS. We performed GWA mapping using a vGWAS package as described by Shen et al.[58] and ~215k SNP markers[59,60]. The significant threshold was determined at a Benjamini and Hochberg false discovery rate level of $q < 0.05$ for correcting multiple testing[61]. For the analysis of YUC8 coding sequences, we downloaded the available coding sequences and predicted amino acid sequences of 139 genome re-sequenced accessions phenotyped in our study from the 1001 Genomes Project (http://signal.salk.edu/atg1001/3.0/gebrowser.php). Sequences of 139 accessions were aligned with ClustalW 2.1 (http://bar.utoronto.ca) to extract SNPs. Only polymorphisms with minor allele frequency (MAF) > 5% were considered. YUC8-based association analysis was performed with a generalized linear model (GLM) implemented in Tassel 2.1[62]. Six significantly associated SNPs according to YUC8-based local association analysis ($P < 0.05$) were taken to define YUC8 haplotypes. Haplogroups containing at least five accessions were used for comparative analysis.

**Plasmid construction and transgenic complementation**. For allelic complementation, we amplified a 1982-bp-long promoter region of YUC8 from genomic DNA of accession Col-0 and the open reading frames carrying the YUC8-hap A or YUC8-hap B allele from Col-0 or Co using the primers listed in Supplementary Data 4, respectively. The amplified fragments were cloned into GreenGate entry modules (pGGA000 for promoter and pGGC000 for open reading frame) and assembled in a pGREEN-IIS-based binary vector following the instructions of Lampropoulos et al.[63]. Plants were transformed through the floral dip method using *Agrobacterium tumefaciens* strain GV3101 containing the helper plasmid pSOUP[64]. Positive transformants were selected on agar plates supplemented with 40 mg L$^{-1}$ hygromycin.

**Histological and fluorescence analyses**. Tissue-specific localization of YUC8 expression was investigated by histological staining of GUS activity in transgenic plants expressing *proYUC8::GUS* described in Hentrich et al.[55]. Root samples were incubated in 20 mg ml$^{-1}$ (w/v) 5-bromo-4 chloro-3-indolyl-β-D-glucuronic acid (X-gluc), 100 mM NaPO$_4$, 0.5 mM K$_3$Fe(CN)$_6$, 0.5 mM K$_4$Fe(CN)$_6$ and 0.1% (v/v) Triton X-100 at 37 °C for 60-90 min in the dark. Samples were then mounted on clearing solution (chloral hydrate: water: glycerol = 8:3:1) for 3 min and imaged using Differential Interference Contrast optics on a light microscope (Axio Imager 2, Zeiss).

For the analysis of cellular traits and expression of fluorophores in LRs, we sampled the four topmost LRs from more than 10 individual plants to minimize developmental stage-dependent variations. Roots were imaged with a laser-scanning confocal microscope (LSM 780, Carl-Zeiss). Excitation and detection of fluorophores were configured as follows: Propidium iodide was excited at 561 nm and detected at 578–718 nm; Venus was excited at 514 nm and detected at 524–540 nm; tdTomato was excited at 561 nm and detected at 566–691 nm. Signal quantifications were performed with ZEN software (Carl-Zeiss).

**Quantitative real-time PCR**. Root tissues were collected by excision and immediately frozen in liquid N. Total RNA was extracted using the RNeasy Plant Mini Kit (Macherey-Nagel GmbH & Co KG, Germany). qRT-PCR reactions were conducted with the CFX 384$^{TM}$ Real-Time System (Bio-Rad, Germany) and the Go Taq qPCR Master Mix SybrGreen I (Promega) using the primers listed in Supplementary Data 4. Relative expression was calculated according to Pfaffl[65] and all genes were normalized to *AtACT2* and *AtUBQ10* as internal references.

**Climate data and statistical analysis**. A subset of climate variables gathered by Hancock et al.[44] was used in this study. Raw data of climate variables (19 climate, latitude and longitude) were downloaded for 113 accessions (Supplementary Data 6) from WorldClim Project (www.Worldclim.org). Climatic variables-SNP associations in Supplementary Fig. 24a, b were extracted from Hancock et al.[44] and replotted.

Root traits in several genotypes were compared by one-way ANOVA followed by post hoc Tukey test at $P < 0.05$. Pairwise comparisons were carried out using Welch's *t*-test. All statistical analyses were performed in R (version 3.6.0)[66].

## Data availability
The authors declare that all data supporting the findings of this study are available within the manuscript and the Supplementary files and provided in the Source Data File. Source data are provided with this paper.

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

## Acknowledgements

We acknowledge the China Scholarship Council (CSC) for financial support to Z.J. (No. 201406350062) and the Deutsche Forschungsgemeinschaft for financial support to N.v.W. (WI1728/25-1) and R.F.H.G. (HE 8362/1-1). We thank Jacqueline Fuge, Annett Bieber, Elis Fraust and Lisa Gruber (Leibniz Institute of Plant Genetics and Crop Plant Research) for excellent technical assistance. We thank Zhaojun Ding (Shandong University, China) for providing seeds of yucQ mutant and Stephan Pollmann (Centro de Biotecnología y Genómica de Plantas (CBGP), Spain) for proYUC8::GUS line. We further thank Thomas Altmann and Rhonda C. Meyer (Leibniz Institute of Plant Genetics and Crop Plant Research, Germany) for providing seeds and SNP data of Arabidopsis accessions.

## Author contributions

Z.J. performed most experiments. R.F.H.G. helped to prepare plasmids and transgenic plants, supervised qRT-PCR analyses, and imaged roots with confocal microscope. Z.J., R.F.H.G., and N.v.W. designed experiments, analyzed the data, and wrote the manuscript. N.v.W. supervised the project.

## Funding

## Competing interests

The authors declare no competing interests.
