## [Peer Review File · Nature Communications]

REVIEWER COMMENTS

Reviewer #1 (Remarks to the Author):

Scope and novelty of research:

Jia et al investigate the foraging response in roots; this is a long-known phenotype that enables roots to preferentially grow in high(er) areas of nitrogen (or other nutrients) in order to overcome the lack of nitrogen in other areas of the root system. The authors find that the enzyme-encoding gene YUCCA8 regulates auxin biosynthesis in lateral root tips, which intersects with a brassinosteroid (BR) signalling module. Crosstalk between these hormonal networks helps control the foraging response and this operates differently in lateral roots compared to the primary root. The authors implicate particular BR signalling components in this intersection. This work is within scope for Nature Communications since it uncovers both new genes and interactions, but also does this by investigating natural variation to identify a new communication hub. This work will help others to approach investigating the foraging response in a new way. It should be of wide interest, not only to plant biologists but to those studying natural variation and those interested in hormonal signalling in any multicellular organism.

Although the discovery that auxin-production/gradients in roots is required for responses to the environment and that the BR and auxin pathways intersect are not novel per se, the discovery that this response is modulated by differences in coding region (rather than expression level or expression regulation) is novel and very interesting. In addition, the bsk3/yuk3 mutant analysis was an elegant way to assess intersection between these pathways, following up with bsk analysis to pinpoint the location of effect to the lateral roots specifically. This makes the findings and their implications novel.

The huge amount of work included was all well-carried out, methods well-compiled, and I found the conclusions to be well-supported. All numerical data was well-displayed and catalogued, with appropriate tests being carried out in order to assess significance. Indeed, the use of a randomised block design for plant growth was a high level of control that is not typically seen.

The putative ER-localisation is interesting as the ER is being increasingly found to act as a hub to shape functions of the cell via affecting responses to the environment and how proteins are stored or mobilised. I expect future work could be done to investigate this at a cell biology level, using tagged YUC8-hap variants.

Major comments:

The only aspect I found to be less well-developed was description of why the 200 accessions were chosen – it was stated that these were chosen as they were diverse (line 66) but what they were diverse in was not described; diversity at the sequence level? Also, at the end, I wanted to know more about the hap A vs. hap B accessions – do these differ in any other phenotypes (from what the authors know)? I could not see anything (by eye) when looking at latitude/longitude data for these, but what about their responses in other known/published studies of root/foraging/nitrogen responses?

Minor comments/typographical

Line 84: deletion of YUC8 – this should be loss of expression of YUC8, not deletion

What is the other significant SNP on Fig.1d – it is not a peak so probably irrelevant, but perhaps worth noting in the legend.

Fig.1d – it would help to label At4g28720 as YUC8, and just use black text rather than the teal colour (or it looks like these should be on chromosome 2).

Figures: What are the black dots/small dashes on the plant seedling images? They are on more, although not all images. They are generally consistent within sets of data, excepting in Supp Fig.7.

Line 201: ‘they were not anymore upregulated by LN’ – change to ‘they were not upregulated by LN as they would be in WT plants’ (since ‘anymore’ could suggest any more, meaning an expression level effect, rather than a yes/no effect)

For some of the supplemental data values I would display the data at the limit of measurable resolution e.g. 2 decimal places for PR values (e.g. Supp Fig.11 data) and 3 decimal places for LR values (e.g. Fig.1 data, Supp Table 3 or Supp Fig.15 data). This does not affect or change any of the figures themselves, or even the data included, just the format it is displayed in.

Reviewer #2 (Remarks to the Author):

Review for Nature Communications

YUCCA-dependent local auxin biosynthesis downstream of brassinosteroids triggers root foraging for nitrogen

Zhongtao Jia, Ricardo F.H. Giehl, Nicolaus von Wirén

Summary

Nitrogen (N) is an essential macronutrient that modulates many plant developmental processes. The development of lateral roots (LR) is strongly determined by nitrogen availability. Plants grown under N deprivation develop longer LRs to reach nitrogen-rich patches. Mechanisms for nitrogen-mediated growth of LR have been reported before. However, there are still many details that remain unclear. In this manuscript, Jia et al. explored how YUCCA8 (YUC8) modulates LR elongation in response to N deficiency. The authors show that N-dependent auxin biosynthesis in LRs acts epistatic and downstream to a canonical brassinosteroid signaling cascade.

Even though the study shows that the methods used are appropriate to identify genes related to signaling pathways, the connection between YUCCA, auxin and N has already been reported. Even though the interplay between BR and auxins in the root response to mild N deprivation is new, the results do not present a major breakthrough in the understanding of the molecular machinery that orchestrates the interplay between N and phytohormones in the control of root system architecture.

Major comments

- The authors do a great job characterizing the root response to mild N availability and the role of YUCCA genes. However, the crosstalk between nitrogen nutrients and different phytohormones in the control of root architecture has been described before. For instance, Yu et al. showed in 2014 that YUCCA and N deprivation are downstream of AGL21. The authors do not refer in detail to the known aspects about N-dependent control of root architecture in both the introduction or discussion. Therefore, these sections seem to be out of context dismissing prior work by several different groups.
- The connection between YUCCAs and mild nitrogen deficiency is undeniable in this work. Authors confirmed a previously reported experiment that established a possible interplay between YUCCA, nitrogen and auxin. Thus, the most novel conclusions defined by the authors is the interplay between BR, auxin, YUCCA and N deprivation (Figure 5). However, while data presented supports a correlation there is not enough evidence to understand cause-consequence connections between these pathways. This reviewer believes additional experiments are required to demonstrate and understand the connection between YUCA and BR in LR length, which is the main novel aspect of this work. There are no insights about how N local perception and signaling is taking control of BR response to regulate LR growth. Is N regulating the expression of any genes associated with BR biosynthesis or signaling in roots in these experimental conditions? Is this a direct or indirect regulation? Can N induce BR accumulation endogenously in roots? Is there any component of N - signaling pathway required for the action of the BR - auxin module?

Specific comments

- Genetic backgrounds are relevant in this study but not clearly described. This information is essential to interpret results. The authors should also clarify if their mutants are knockdown or true knockout lines. It is recommended to add a Supplementary Table to organize all this information including all mutants.
- 200 different accessions are used to determine natural variation of LR length to N availability but you only show 9 different ecotypes to conclude that there is no relationship between LR response and YUC8 expression. I understand that you have representative natural accessions with contrasting N responses, how many of these 9 ecotypes are representing mild, medium and high responses to low N? Why not including Par-3, which present the highest increase in LR length? At least for Co, there can be a relationship between root response and YUC8 expression. It is also not clear why the authors decided to search SNPs in 139 accessions and not the 200 initially used. The authors should explain this briefly.
- l.212: You conclude that "LN-induced LR elongation relies on BR signaling-dependent stimulation of YUC5/7/8 expression to increase local auxin levels", however you showed that there was no relationship between YUC expression or change in expression in response to LN and root responses. How is then this conclusion supported?

- Is this YUC-dependent LR elongation phenotype dependent on a specific source of N, or is specifically connected to nitrate deficiency?
- It is widely reported that N-elicited root architecture modifications are mediated by nitrogen and phytohormone crosstalk. For instance, nitrate interplay with abscisic acid, auxin, and cytokinin has been reported to locally and systemically control root architecture. The authors did not discuss this topic in the introduction. The introduction needs to be improved by citing all relevant prior work.
- T-Allele is mentioned to be associated with longer LR in response to LN. However, despite being statistically significant, the difference shown in Supplementary Figure 1a between A and T-Alleles under an LN regime is almost negligible. Authors should show representative images of accessions corresponding to A and T-alleles to evidence the identified phenotype's relevance.
- In Figure 1d the authors state that they detected one prominent peak associated with the LR response to low nitrogen on chromosome 4. Nevertheless, it can be observed that another peak is present on chromosome 4 which was omitted by the authors. In this respect, the authors should clarify why they only studied one peak and not the other.
- The authors showed that the identified SNP was located in YUC8 gene. In this regard, they analyzed T-DNA insertion lines of YUC8 and two other genes located within the 20-kb interval centered around the identified SNP. The authors observed no significant statistical differences for average LR length between Col-0 and T-DNA insertional lines for At4g28730 and At4g28740. Nevertheless, they only evaluated one mutant allele per gene. The authors should also evaluate the response of other mutant alleles for the studied genes in order to have a more robust result. In addition, the authors should verify through real-time qRT-PCR that the T-DNA insertional mutants show a decrease on the gene's transcripts when compared to Col-0 plants.
- The authors evaluated root architectural traits for YUC8 closest homologs in root: YUC3, YUC5, and YUC7. For this purpose, they used single mutants. Nevertheless, and as stated above, the authors should evaluate more than one mutant allele for each gene in order to have more robust results. In addition, the yuc3 (GK-376G12) mutant has the T-DNA insertion in an intron. Thus, the authors must complement the root phenotypic analysis with a qRT-PCR in order to assess if the mutant lines actually show a decrease in the gene's transcript when compared to control plants.
- In Supplementary Fig.3, the authors observed that yuc3, yuc5, and yuc7 mutants show a decrease in primary root and LR lengths under N deficiency. However, the yuc-5 mutant has a Col-3 background and the authors compare it with Col-0. Thus, the authors must compare the response of the yuc-5 mutant line with a Col-3 line.
- Supplementary Table 1 with information about the accessions used in this study is missing and Supplementary Table 2 containing information about SNPs in YUC8 coding sequence are missing. Supplementary Table 3 with information about the two major haplotypes is also lacking. Supplementary Table 4 with the primers used in this study is missing as well.
- Line 149 is not finished. The position of the substitution must be stated.
- The authors should report the experiments by which they confirmed yucQ transformation with the YUC8-hap A and YUC8-hap B haplotypes.
- The authors transformed yucQ mutants with a Col-0 promoter in Figure 3. However, they did not specify the yucQ background. If this mutant did not have a Col-0 background, how can the authors be sure that phenotypes observed in yucQ and yucQ complemented lines are comparable with Col-0? The yucQ background needs to be specified.
- In figure 3e-f, authors compared PR and LR length between mutant and complemented genotypes. The

authors concluded that YUC8-hapA variant rescued more efficiently PR length. However, the numbers used in these experiments are not comparable. Despite agreeing with the mentioned conclusions, we suggest performing a statistical analysis with a comparable N number.

- In Figure 4 (a-c), the letters showing statistical differences are too big and the box plots too small, making it difficult to read the graphs. The same can be said for Figure 4 e-f; Uppercase letters should be placed next to the boxplot. The graphs in this Figure must be improved. The letters showing statistical differences in Fig. 5 b and c are too big when compared to the boxplots. The authors should improve the format of the graphs.

- Most of the discussion is focused on explaining why the YUC8-Hap A is associated with LR growth under a low N concentration, and states the novelty of this study. However, the authors poorly contrast their results with the known signaling pathways, molecular components and already proposed models of N signaling underlying root responses. Are there any N- responsive transcription factors that could be candidates for the control of BR - associated genes? Are these results consistent with any other models of auxin - mediated control of root architecture in response to N availability? How can the study's results be incorporated into these already proposed models?

- Line 66-67: Plants are not grown in high or low N for 9 days. Plants are grown for one week on sufficient N, plus 9 days on HN or LN. Please change to make it more clear to the reader.

- Fig 1c: This figure is not showing relative average LR length, as indicated in the legend. Please change "relative average LR length" for an x-axis title "LN/HN LR length ratio"

- Line 82: "we nonetheless analyzed" I think this conjunction is not well used in this context. Maybe change to "we then analyzed..."

- Supp Fig. 2ab: There are no boxes indicating what the colors mean.

- Supp Fig. 2A should be included in Fig 1, since it is part of the root morphological analysis of the WT vs. YUC8 mutants. Total root length graph is redundant since there is already a plot showing PR length and LR length.

- Figure 2: You have already shown a LR phenotype for the yuc8 mutants, why center the cell elongation analysis on a mutant that includes genes that are not expressed in roots? The yucQ mutant is associated with an important decrease of IAA content that can mask a more-specific, YUC8-N-related phenotype.

- Line 112: "Conversely, when YUCCA-dependent auxin biosynthesis in roots of wild-type plants was suppressed with 4-phenoxyphenylboronic acid (PPBo)..." Since PPBo is an inhibitor of YUCCA activity, how can you be sure that auxin biosynthesis is specifically suppressed in the roots?

- Supp Fig. 8. What happens with PR length response to low N? Is it also altered by NPA (I can't really tell from the photograph)? A graph of PR root length should also be added to Supp Fig 8 to support the conclusion that "these results collectively indicate that YUCCA-mediated local auxin production in roots modulates root elongation under mild N deficiency"

- This paragraph (l. 128-137) should be included before the analysis of gene expression, to support the conclusion that YUCCA-mediated local auxin production modulates root elongation under mild N deficiency (l.118-119).

- Supp Fig. 9: Please include panel a) on Fig 2e

Reviewer #3 (Remarks to the Author):

In this manuscript, Jia et al. takes advantage of different Arabidopsis accessions carrying natural variation to evaluate their root response to nitrogen deficiency, intending to seek the components involved in this biological process. Using GWAS and SNP analysis, the authors identified a target gene YUC8, an auxin biosynthesis-related gene, and proposed that this gene's family members contribute to lateral root elongation response to N deficiency. The genetic experiments using yuc single or multi-mutants further confirmed that the YUC-mediated local auxin biosynthesis process in root is associated with the root response to nitrogen. Consistently, the YUC genes' expression levels could be triggered by N deficiency. Meanwhile, the authors identified two allelic variants of YUC8 harboring a non-synonymous substitution of one amino acid. They demonstrated that the different biological activity between YUC8 variants underlies the differential lateral root foraging for nitrogen for these Arabidopsis accessions. Moreover, the genetic combined with the pharmacological experiments imply the auxin signaling pathway is downstream of the BR signaling in the root foraging for nitrogen.

Generally, this work seems interesting, especially the crosstalk signaling between plant hormones brassinosteroid and auxin in root foraging for N. The data are also well organized and the paper is very easy to be followed. However, some of the conclusions drawn from the data are not so persuasive, and I suggest the authors revise the text or add much more data to make it more explicit and accurate. Meanwhile, some analyses are a bit superficial. I have some comments below for authors to improve.

Major comment

1. The information conveyed by this title is not exactly what the data can support. The GWAS analysis results only conclude that the allelic variants of YUC8 contribute to the differential local auxin biosynthesis activities among these Arabidopsis accessions and thus trigger their different root foraging for nitrogen, but it can not be used as the regulation mechanism in plants as title proposed. concerning the mechanism, from the existing data, I believe that the auxin biosynthesis triggers root foraging for nitrogen, but no data here shows that only YUC genes are up-regulated in the mild N deficiency. Are the expression levels of other crucial auxin biosynthesis-related genes (e.g. TAA) up-regulated under nitrogen deficiency? Do their corresponding chemical inhibitors (e.g. L-kyn inhibits TAA activity) also disturb the lateral root elongation in low nitrogen as YUC inhibitor PPBo shows? If yes, it should be very careful to use the "YUC-dependent local auxin biosynthesis", because other auxin synthesis genes are also involved in this process.

Therefore, more experiments should be considered to enhance the view as the title proposed or modify the title.

2. Usually, the genetic variations of geographically representative Arabidopsis accessions confer their fitness in a certain habitat. However, I did not see any this kind of analysis throughout the paper. To go much deeper, I urge the authors to further check the geographical distribution between the two groups of Arabidopsis accessions carrying the YUC8 variant YUC8-hapA and YUC8-hapB separately. Then combined with the feature of the local environments, it might provide some insights regarding how the auxin-dependent root foraging for N is modified by natural selection and linked to their adaptability to local environments. At least, it should be discussed in the discussion section.

3. Performing experimental analysis with only Arabidopsis accession Col-0 to verify the GWAS results is not so convincing. If some other Arabidopsis accessions (e.g. Co) can be used for similar experimental analysis, the conclusions would be much more solid.

For example, even though the bioinformatic analysis (GWAS, SNP) proposed that YUC8 coding variants (hapA and hapB) link to the diverse root response to N among these Arabidopsis accessions. However,

no further data demonstrate if the differential transcriptional level of YUC8 is also involved in the diverse nitrogen responses or not. Even though to request some transgenic work is not so reasonable, some other simple experiments can be performed using other Arabidopsis accessions. Is the transcription level of YUC8-hapB in accession Co also up-regulated under low N condition? Is its increased expression level of YUC8-hapB comparable to that of YUC8-hapA in Col background? Moreover, to further prove that the different auxin accumulations of these Arabidopsis accessions lead to their varied root response to nitrogen, the exogenous 50nM IAA can be applied in different Arabidopsis accessions, to see if the elimination of endogenous auxin accumulation differences would abolish the differences of root N response among these Arabidopsis accessions.

4. Something needs to be further clarified. The author assessed 200 Arabidopsis accessions and then identify YUC8 is involved in the diverse LR response to low N among these accessions. However, to identify the SNPs within YUC8 coding sequence, they used 139 Arabidopsis accessions. It is better to explain why different numbers of accessions are used for these two analyses and what criteria they applied for selecting the accessions to perform these bioinformatic analyses.

Minor comment

1) Some descriptions are a bit overstated in the Abstract section. Page 1 line 11, it should be the “flowering plants”, not the “plants”. Line 32, this work only implies the ‘root response to nitrogen availability’ but not the ‘root response to abiotic stress’.

2) The concentration of these chemicals used for plant treatment should be clearly showed. It is better to mention them when they appeared in the main text for the first time. For example, it is very hard for me to find which concentration of NPA they used in both the main text and figure legends.

3) Line 229, it is hard to have the conclusion that “mild N deficiency stimulated local auxin biosynthesis in the root apical meristem”. Firstly, no evidence shows that the auxin biosynthesis genes (e.g. YUC) are only up-regulated in root apical meristem. Secondly, even though the R2D2 signal is enhanced in RAM, however, it is also possible that the excessive synthesized auxin in other tissue is transported to RAM. It is much accurate to say the “local auxin accumulation in the root meristem”

4) Line 274, there are few minor grammar mistakes, check through the manuscript to correct them. e.g. Line 274, ‘were’ should be changed to ‘was’. Line 149, some words seem to be missing after ‘at’.

5) Some words are very hard to be digested. ‘Homozygosity’ should be replaced by ‘homozygote’.

Reviewer #4 (Remarks to the Author):

The manuscript of Jai et al describes the natural genetic variation in YUC8 is involved in auxin biosynthesis as cause for the root foraging response to low N. Overall, manuscript is well written, authors have done a great job by doing many detailed and in-depth experiments to support their findings and conclusions.

However, I have a number of questions/comments regarding the data presented and there are still some issues that authors should take into account.

1) Authors only focused on one GWA peak located on chromosome 4 where top marker SNP located

close to YUCCA8 gene. However, GWA analysis clearly also identified other regions with significant association (on Chr. 4 and Chr. 2). Even though further experiments carried out meticulously by authors to show YUC8 as determinant for LR response to low N. In order rule out bias approach to choose YUC8 region for follow up experiments, authors should also investigate whether other genes located close to top SNPs from other two significant associations (shown in Fig. 1d.) are contributing to the LR response to LN or not. Authors need to perform expression analysis under HN/LN and phenotyping using T-DNA insertion lines of the genes falls within 10 to 20 kb regions from top SNPs present on Chr. 2 and Chr. 4.

2) Results section is very well written but introduction and discussion sections are too short. Authors should rewrite both the sections. Authors can further enrich the introduction part by giving better background on the LR development and N availability. For example: adding “role of TAR2 in low nitrogen-mediated reprogramming of root architecture in Arabidopsis” etc etc.

3) Authors should also discuss Auxin-BR crosstalk in more detail instead of writing just few lines on this interesting aspect. Also, in this case authors can prepare a schematic illustration describing the crosstalk between auxin-BR signaling and its role in LR growth under LN.

4) Figure 4 seems very complex, I would strongly suggest authors to replot this figure in a different way for better understanding for the readers.

5) In Methodology section sufficient information is not given, for example haplotype analysis part is too short. Also, in many experiments sample size varies a lot and in some cases number of individuals are not optimum in my opinion. It is important to mention in that case whether experiments have been repeated multiple times or not.

6) Authors last paragraph (Line 267-270) in discussion section is superficial and overclaiming.

REVIEWER COMMENTS

Reviewer #1 (Remarks to the Author):

Scope and novelty of research:

Jia et al investigate the foraging response in roots; this is a long-known phenotype that enables roots to preferentially grow in high(er) areas of nitrogen (or other nutrients) in order to overcome the lack of nitrogen in other areas of the root system. The authors find that the enzyme-encoding gene YUCCA8 regulates auxin biosynthesis in lateral root tips, which intersects with a brassinosteroid (BR) signalling module. Crosstalk between these hormonal networks helps control the foraging response and this operates differently in lateral roots compared to the primary root. The authors implicate particular BR signalling components in this intersection. This work is within scope for Nature Communications since it uncovers both new genes and interactions, but also does this by investigating natural variation to identify a new communication hub. This work will help others to approach investigating the foraging response in a new way. It should be of wide interest, not only to plant biologists but to those studying natural variation and those interested in hormonal signaling in any multicellular organism. Although the discovery that auxin-production/gradients in roots is required for responses to the environment and that the BR and auxin pathways intersect are not novel per se, the discovery that this response is modulated by differences in coding region (rather than expression level or expression regulation) is novel and very interesting. In addition, the *bsk3/yuk3* mutant analysis was an elegant way to assess intersection between these pathways, following up with *bsk* analysis to pinpoint the location of effect to the lateral roots specifically. This makes the findings and their implications novel. The huge amount of work included was all well-carried out, methods well-compiled, and I found the conclusions to be well-supported. All numerical data was well-displayed and catalogued, with appropriate tests being carried out in order to assess significance. Indeed, the use of a randomised block design for plant growth was a high level of control that is not typically seen.

Response: Thanks for the encouraging comments about our study.

The putative ER-localisation is interesting as the ER is being increasingly found to act as a hub to shape functions of the cell via affecting responses to the environment and how proteins are stored or mobilised. I expect future work could be done to investigate this at a cell biology level, using tagged YUC8-hap variants.

Response: Thanks for this interesting suggestion. Indeed, we are planning to continue research on the identified YUC8 variants, which will include intracellular localization studies.

Major comments:

The only aspect I found to be less well-developed was description of why the 200 accessions were chosen – it was stated that these were chosen as they were diverse (line 66) but what they were diverse in was not described; diversity at the sequence level?

Response: We apologize for not being clear. In our institute, a genetic diverse collection of 200 *Arabidopsis thaliana* accessions has been assembled, which is based on geographical distribution (as described in Jia et al., 2019, Nature Commun.) and which have been genotyped with a 250 k chip. To clarify this point in the manuscript, we rephrased the text to “in a geographically and genetic diverse panel of 200 *A. thaliana* accessions...” (Line 77).

Also, at the end, I wanted to know more about the hap A vs. hap B accessions – do these differ in any other phenotypes (from what the authors know)?

Response: While scoring lateral root length for these 200 lines, we also assessed other root traits such as primary root length, total lateral root length, total root length and lateral root number under HN and LN conditions. In addition to longer average lateral root length, YUC8-Hap A accessions had

on average also longer total root length and total lateral root length (which are related parameters) under low N than YUC8-Hap B accessions. However, there was no significant difference for primary root length and lateral root number. There was also no difference in the shoot phenotype or in shoot biomass. We now included this piece of information in Supplementary Fig. 16 and in the revised text (*Lines 196-197*).

I could not see anything (by eye) when looking at latitude/longitude data for these, but what about their responses in other known/published studies of root/foraging/nitrogen responses?

Response: To address this question, we checked the response of the contrasting lines identified here in other GWAS reports presenting phenotypical data for root traits (Gifford et al., 2013, Plos Genet., Satbhai et al., 2017, Nat. Commun.; Li et al., 2019, Nat. Commun.; Ogura et al., 2019, Cell). However, with regard to root morphological responses to Fe deficiency/toxicity, nitrate starvation or auxin transport inhibition by NPA, we did not find substantial genotypic differences for our contrasting lines.

Minor comments/typographical

Line 84: deletion of YUC8 – this should be loss of expression of YUC8, not deletion

Response: Thanks. We now rephrased the text to “loss of *YUC8* expression” (*Line 102*) in the revised manuscript.

What is the other significant SNP on Fig.1d – it is not a peak so probably irrelevant, but perhaps worth noting in the legend.

Response: Actually, we did not consider it further as the peak consisted only of one SNP. But we will check it again. We now noted in the revised manuscript that we chose the peak with further supporting SNPs (*Lines 86-91*).

Fig.1d – it would help to label At4g28720 as YUC8, and just use black text rather than the teal colour (or it looks like these should be on chromosome 2).

Response: Thanks for pointing to this. We followed this suggestion and revised Fig. 1d.

Figures: What are the black dots/small dashes on the plant seedling images? They are on more, although not all images. They are generally consistent within sets of data, excepting in Supp Fig.7.

Response: The black dashes/dots represent the position of the primary root tip at the time when the seedlings were transferred from the pre-culture to the N treatments. We now added this information in the Material and Methods (*Lines 387-389*).

Line 201: ‘they were not anymore upregulated by LN’ – change to ‘they were not upregulated by LN as they would be in WT plants’ (since ‘anymore’ could suggest any more, meaning an expression level effect, rather than a yes/no effect)

Response: Thanks, done!

For some of the supplemental data values I would display the data at the limit of measurable resolution e.g. 2 decimal places for PR values (e.g. Supp Fig.11 data) and 3 decimal places for LR values (e.g. Fig.1 data, Supp Table 3 or Supp Fig.15 data). This does not affect or change any of the figures themselves, or even the data included, just the format it is displayed in.

Response: Thanks, done!

Reviewer #2 (Remarks to the Author):

Review for Nature Communications

YUCCA-dependent local auxin biosynthesis downstream of brassinosteroids triggers root foraging for nitrogen

Zhongtao Jia, Ricardo F.H. Giehl, Nicolaus von Wirén

Summary

Nitrogen (N) is an essential macronutrient that modulates many plant developmental processes. The development of lateral roots (LR) is strongly determined by nitrogen availability. Plants grown under N deprivation develop longer LRs to reach nitrogen-rich patches. Mechanisms for nitrogen-mediated growth of LR have been reported before. However, there are still many details that remain unclear. In this manuscript, Jia et al. explored how YUCCA8 (YUC8) modulates LR elongation in response to N deficiency. The authors show that N-dependent auxin biosynthesis in LRs acts epistatic and downstream to a canonical brassinosteroid signaling cascade. Even though the study shows that the methods used are appropriate to identify genes related to signaling pathways, the connection between YUCCA, auxin and N has already been reported.

Response: Indeed, previous studies have reported that auxin levels increase in roots of low N - exposed plants (Caba et al., 2000, *Planta*; Walch-Liu et al., 2006, *Ann. Bot.*; Tian et al., 2008, *J. Plant Physiol*; Kiba et al., 2011, *J. Exp. Bot.*). In part, these studies even suggested that the higher auxin levels detected in N-deficient roots originate from increased shoot-to-root auxin transport (Forde, et al., 2002; *J. Exp. Bot.*; Kiba et al., 2011, *J. Exp. Bot.*). However, all these observations remained at the physiological level. One exception is the study of Ma et al. (2013, *Plant J*) showing that TAR2-dependent local auxin biosynthesis is induced by mild N deficiency. However, in *tar2* mutant plants lateral root emergence but not lateral root elongation is impaired. To the best of our knowledge, our findings that YUCCA proteins stimulate local auxin biosynthesis in response to mild N deficiency and that this activity has a specific effect on lateral root elongation is completely novel (see additional comments below).

Even though the interplay between BR and auxins in the root response to mild N deprivation is new, the results do not present a major breakthrough in the understanding of the molecular machinery that orchestrates the interplay between N and phytohormones in the control of root system architecture.

Response: We disagree with the reviewer's view. The synergistic action of BRs and auxin in lateral root elongation was not documented before. Regarding this N deficiency response, we could even place auxin biosynthesis downstream of BR signaling, which is not the case for most other studies having reported on BR-auxin crosstalks that primarily involves auxin signaling and transport (Nakamura et al., 2003, *Plant Physiol.*; Bao et al., 2004, *Plant Physiol.*; Nemhauser et al., 2004, *Plos Biol.*; Mouchel et al., 2006, *Nature*; Vert et al., 2008, *PNAS*; Zhou et al., 2013, *Mol. Plant*; Cho et al., 2013, *Nat. Cell Biol.*; Retzer et al., 2019, *Nat. Commun.*). Moreover, our study demonstrates that YUC-dependent auxin biosynthesis represents a novel key element in this signaling cascade, which together with our previous findings (Jia et al., 2019, *Nat. Commun.*; Jia et al., 2020, *Plant Physiol.*) now shows that *in-situ* biosynthesis of BR and auxin determine N-dependent root elongation. This advances our understanding relative to previous studies, which proposed that polar auxin transport was key for N-dependent changes in root morphology (Forde, et al., 2002, *J. Exp. Bot.*; Kiba et al., 2011, *J. Exp. Bot.*). Beyond that, our study shows the existence of natural variation for YUC8-driven auxin biosynthesis which greatly impacts on the adaptation of root traits to mild low N. The uncovered N-dependent, YUC-mediated local auxin biosynthesis in roots downstream of BR signaling represents a novel regulatory module and thus a significant advance in the mechanistic understanding of root plasticity changes in response to environmental cues.

Major comments

- The authors do a great job characterizing the root response to mild N availability and the role of YUCCA genes. However, the crosstalk between nitrogen nutrients and different phytohormones in the control of root architecture has been described before. For instance, Yu et al. showed in 2014 that YUCCA and N deprivation are downstream of AGL21. The authors do not refer in detail to the known aspects about N-dependent control of root architecture in both the introduction or discussion. Therefore, these sections seem to be out of context dismissing prior work by several different groups.

Response: We regret that we did not detail all prior work. Indeed, we agree with the reviewer that the study of Yu et al. (2014) is relevant for our story. Therefore, we now included it in the revised Introduction (*Lines 53-54*) and Results (*Lines 147-158*). In addition, we have extensively revised the Introduction and Discussion sections to address the reviewer's concerns.

- The connection between YUCCAs and mild nitrogen deficiency is undeniable in this work. Authors confirmed a previously reported experiment that established a possible interplay between YUCCA, nitrogen and auxin.

Response: We disagree with the reviewer that the connection between YUCCA-dependent auxin biosynthesis and mild N deficiency is just a confirmatory finding. To the best of our knowledge, previous studies did not show if and which YUCCAs are responsive to N deficiency nor did they address which YUCCAs are critical for root architectural modifications induced by low N availability. Besides addressing these aspects with independent approaches, we also identified natural YUC8 haplotypes that drive differential N-dependent lateral root elongation at the species level. Therefore, we believe that our results extend our knowledge well beyond what was known before.

Maybe, the reviewer's comment refers to the study of Yu et al. (2014, Mol. Plant), which showed that the transcription factor AGL21 regulates the expression of some YUCCA genes and root auxin accumulation, and demonstrate that AGL21 modulates lateral root growth under a N-free condition. However, this study did not investigate if the expression of YUCCA genes is induced by N starvation nor if *yucca* mutants show altered lateral root growth in response to N deprivation. Furthermore, the phenotypes of *agl21* mutant and AGL21 overexpressing lines were recorded under N-free, i.e. severe N deficiency, which induces distinct root architectural changes compared to the mild N deficiency that was investigated here. However, we took up the reviewer's comment and investigated whether AGL21 and its paralogous gene ANR1 are required to stimulate root elongation in response to mild N deficiency by using the *agl21 anr1* double mutant described previously (Liu et al., 2020, J. Exp. Bot.). As shown in new Supplementary Fig. 11, we observed that the *agl21 anr1* double mutant showed a similar response to mild N deficiency as wild-type plants. Furthermore, we phenotyped the *yuc8* mutants and the *yucQ* mutant under severe N deficiency, a condition similar to that used by Yu et al. (2014, Mol. Plant). The length of primary root and lateral roots of all tested mutants was not significantly different to wild type (new Supplementary Fig. 12 and 13). Taken together, these results show that distinct signaling pathways are involved in root adaptive responses to mild and severe N limitations. These new results and the reference to the study of Yu et al. (2014) are now presented in *Lines 147-158*.

Thus, the most novel conclusions defined by the authors is the interplay between BR, auxin, YUCCA and N deprivation (Figure 5). However, while data presented supports a correlation there is not enough evidence to understand cause-consequence connections between these pathways. This reviewer believes additional experiments are required to demonstrate and understand the connection between YUCA and BR in LR length, which is the main novel aspect of this work. There are no insights about how N local perception and signaling is taking control of BR response to regulate LR

growth. Is N regulating the expression of any genes associated with BR biosynthesis or signaling in roots in these experimental conditions?

Response: As mentioned in the Introduction, our previous studies have shown that mild N deficiency systemically up-regulates the expression of critical BR biosynthesis genes, including *DWF1*, *CPD*, *DWF4* and *BR6OX2*, and induces the expression of the BR co-receptor *BAK1* (Jia et al., 2019, Nat. Commun.; Jia et al., 2020, Plant Physiol.).

Is this a direct or indirect regulation?

Response: We expect that mild N deficiency-induced biosynthesis of BR via *DWF1*, *CPD*, *DWF4* and *BR6OX2* is indirect, most likely mediated by an N-responsive transcription factor, but we don't know yet which one. Identification of such transcription factor(s) is beyond the scope of the present study.

Can N induce BR accumulation endogenously in roots?

Response: Considering the difficulty to detect BRs in roots from agar-grown *Arabidopsis* plants, in our previous study (Jia et al., 2020, Plant Physiol.), we assessed the transcript levels of *DWF1*, *CPD*, *DWF4* and *BR6OX2* commonly used as molecular marker of BR levels (Shimada, et al., 2003, Plant Physiol.; Kim et al., 2006, Plant Physiol.; Chung et al., 2011, Plant J; Poppenberger et al., 2013, EMBO J; Singh et al., 2014, Plant Physiol.; Martins et al., 2017, Nat. Commun.; Wei et al., 2017, Plant Physiol.). As these genes were up-regulated by mild N deficiency (Jia et al., 2020, Plant Physiol.), it is expected that endogenous BR accumulation in roots is indeed increased in response to mild N deficiency.

Is there any component of N - signaling pathway required for the action of the BR - auxin module?

Response: Given that the various components of the uncovered local BR-auxin module in roots are controlled by the plant's shoot N status, we expect that some of the known components of systemic N signaling are indeed involved in this regulation. For instance, lateral root elongation is stimulated in *cepr1* and *cepr1/2* mutants (Tabaka et al., 2014, Science), suggesting that systemic N signaling via the CEP-CEPRs-CEPDs cascade may be involved. We now discuss this point in the revised manuscript (Lines 350-353).

Specific comments

- Genetic backgrounds are relevant in this study but not clearly described. This information is essential to interpret results.

Response: Thanks for pointing this out. We carefully checked our mutant lines and found all mutants were in the background of Col-0, except for *yuc5-1* (SAIL_116_C01) which is in the background of Col-3. We clarified these details in the Methods (Line 361) and in the figure legends as well.

The authors should also clarify if their mutants are knockdown or true knockout lines. It is recommended to add a Supplementary Table to organize all this information including all mutants.

Response: Thanks for this suggestion. The *yuc8-1* (SALK_096110C) and *dwf4-44* (SAIL_882_F07) have been characterized and documented as knockdown mutant in Sun et al. (2012, Plos Genet.) and in Du et al. (2017, Front. Plant Sci.). For all other mutants for which no expression data was reported previously, we performed qRT-PCR in the revised manuscript to verify the expression level of the targeted gene in each mutant line. The results are presented in the new Supplementary Fig. 25. We furthermore followed the reviewer's suggestion and organized the relevant information for each mutant in the new Supplementary Table 5.

- 200 different accessions are used to determine natural variation of LR length to N availability but you only show 9 different ecotypes to conclude that there is no relationship between LR response and YUC8 expression. I understand that you have representative natural accessions with contrasting

N responses, how many of these 9 ecotypes are representing mild, medium and high responses to low N?

Response: Among the selected accessions, Co, Ty-0 and Edi-0 are weak responders (increase of average lateral root length under low N relative to high N of less than 40%), Col-0 and Ven-1 intermediate responders (increases of 50% and 80%, respectively), while Kas-2, JEA, Tamm-27 and Uod-1 represent strong responders (increases of more than 100%).

Why not including Par-3, which present the highest increase in LR length?

Response: We did not include Par-3 as we did not have sufficient amounts of well-germinating seeds at the time when the experiment was carried out. However, we included Uod-1, an accession that showed a 185% increase in average lateral root length in low N compared to high N, which was very similar to the 188% increase shown by Par-3. For clarification, we now added the percent increase of lateral root elongation (LN-to-HN ratio) on top of the bars in the revised Supplementary Fig. 15a (Supplementary Fig. 10a of previous version of manuscript).

At least for Co, there can be a relationship between root response and YUC8 expression.

Response: We would like to point out that interpreting the data based on a single genotype is not meaningful to establish a relation for the observed intraspecific variation. Instead, *YUC8*-based association mapping and subsequent haplotype analysis were helpful in elucidating the genetic basis contributing to the natural variation. In a second step, we then compared *YUC8* expression in different natural accessions belonging to different haplogroups.

It is also not clear why the authors decided to search SNPs in 139 accessions and not the 200 initially used. The authors should explain this briefly.

Response: We did not apply any criterion to select 139 accessions for *YUC8*-based association analysis. In the GWAS analysis, we phenotyped 200 accessions that are genotyped with a 250K chip (Atwell et al., 2010, Nature; Horton et al., 2012, Nat. Genet.). However, of these 200 lines, complete genomic information is available for only 139 accessions which have been re-sequenced (1001 Genomes Project; <http://signal.salk.edu/atg1001/3.0/gebrowser.php>). This point is now clarified in the text (Lines 411-414).

- I.212: You conclude that “LN-induced LR elongation relies on BR signaling-dependent stimulation of YUC5/7/8 expression to increase local auxin levels”, however you showed that there was no relationship between YUC expression or change in expression in response to LN and root responses. How is then this conclusion supported?

Response: This conclusion is based on the results we obtained with BSK- and BZR1-dependent expression of *YUC8* and other *YUCs* (Fig. 5f, g; Supplementary Fig. 23 of the revised manuscript), the auxin reporter expression in the *bzr1-1D* background (Fig. 5h-k) and the chemical complementation of the *bsk* mutants by external IAA (Fig. 5e). The point made by the reviewer may refer specifically to the variation of the lateral root foraging response among natural accessions, where no correlation between *YUC8* expression and lateral root response to low N was observed. In this context, allelic variation in the coding sequence of *YUC8* is more relevant to explain phenotypic differences between accessions, as also confirmed by our transgenic allelic complementation experiments shown in Figs. 3 and 4.

- Is this YUC-dependent LR elongation phenotype dependent on a specific source of N, or is specifically connected to nitrate deficiency?

Response: As we did not intend to assess the effect of nitrate or ammonium but rather to decipher the mechanism underlying changes in lateral root elongation in response to external N levels previously determined to induce mild deficiency in shoots (Gruber et al., 2013 *Plant Physiol.*), we supplied N as NH_4NO_3 and KNO_3 and lowered the supplied concentration from 11.4 mM to 0.55 mM without altering the ratio between NH_4NO_3 and KNO_3 (i.e., 1:9.4 in both cases).

- It is widely reported that N-elicited root architecture modifications are mediated by nitrogen and phytohormone crosstalk. For instance, nitrate interplay with abscisic acid, auxin, and cytokinin has been reported to locally and systemically control root architecture. The authors did not discuss this topic in the introduction. The introduction needs to be improved by citing all relevant prior work.

Response: Thanks for pointing this out. We have now substantially revised the Introduction to refer to aspects that have been already reported by previous studies. *Lines 42-44 and 51-54.*

- T-Allele is mentioned to be associated with longer LR in response to LN. However, despite being statistically significant, the difference shown in Supplementary Figure 1a between A and T-Alleles under an LN regime is almost negligible.

Response: As root architectural traits including root length are quantitative traits under the control of multiple loci with minor to moderate effects, it is not surprising that the difference between two alleles is not very large. This has been exemplified in many studies (Lachowiec et al., 2015, *Plos Genet.*; Kobayashi et al., 2016, *Plant Cell & Environ.*; Tang et al., 2018, *Nat. Commun.*; Li et al., 2019, *Nat. Commun.*; Ogura et al., 2019, *Cell*; Jia et al., 2019, *Nat. Commun.*), where the detected loci contributed only 10% to 20% of the observed phenotypic variation for a given quantitative trait.

Authors should show representative images of accessions corresponding to A and T-alleles to evidence the identified phenotype's relevance.

Response: Thank you for this suggestion. We now included this information in revised Fig. 1a.

- In Figure 1d the authors state that they detected one prominent peak associated with the LR response to low nitrogen on chromosome 4. Nevertheless, it can be observed that another peak is present on chromosome 4 which was omitted by the authors. In this respect, the authors should clarify why they only studied one peak and not the other.

Response: Actually, we chose this SNP because 1) it is significantly associated with LR response, 2) is supported by adjacent markers, and 3) T-DNA insertion lines were available for all genes falling within the 20-kb supporting interval, allowing us to determine experimentally which of the genes sitting within this interval was the most promising candidate for further investigation. The outcome of our experiments enabled us to establish a significant impact of YUC8 on the investigated response. We followed the suggestion of Reviewer 1 by mentioning the other significant peak in *Lines 86-91.*

- The authors showed that the identified SNP was located in YUC8 gene. In this regard, they analyzed T-DNA insertion lines of YUC8 and two other genes located within the 20-kb interval centered around the identified SNP. The authors observed no significant statistical differences for average LR length between Col-0 and T-DNA insertional lines for At4g28730 and At4g28740. Nevertheless, they only evaluated one mutant allele per gene. The authors should also evaluate the response of other mutant alleles for the studied genes in order to have a more robust result. In addition, the authors should verify through real-time qRT-PCR that the T-DNA insertional mutants show a decrease on the gene's transcripts when compared to Col-0 plants.

Response: Due to limited time for revision, we were not able to isolate a second homozygous mutant allele for At4g28730 and At4g28740. However, we could verify by qRT-PCR that the T-DNA insertion

lines for *At4g28730* (SALK_077059C) and *At4g28740* (SAIL_1286_E04C) represent real knock-outs (new Supplementary Fig. 25a,b). In addition, we found that although both genes are expressed in roots, none of them is up-regulated by N deficiency (Supplementary Fig. 1, new panels d and e), providing additional evidence that they unlikely contribute to root elongation under low N. Moreover, their functions are annotated as “iron-sulfur cluster assembly” and “chloroplast organization”, which have so far no obvious relation to N-dependent root elongation.

The authors evaluated root architectural traits for YUC8 closest homologs in root: YUC3, YUC5, and YUC7. For this purpose, they used single mutants. Nevertheless, and as stated above, the authors should evaluate more than one mutant allele for each gene in order to have more robust results.

Response: In the revised manuscript, we confirmed by qRT-PCR that *yuc5-1* (SAIL_116_C01) and *yuc7-1* (SALK_059832C) represent real knockouts (new Supplementary Fig. 25d and f). In addition, we isolated the new knockout line *yuc5-2* (SALK_088618C) and *yuc7-2* (SALK_034074C). We phenotyped *yuc5-2* and *yuc7-2* under mild N deficiency and found that lengths of primary root as well as lateral roots were significantly shorter than those of wild-type Col-0 plants (new Supplementary Fig. 3 and 4), validating the phenotypes previously recorded for *yuc5-1* and *yuc7-1*.

In addition, the *yuc3* (GK-376G12) mutant has the T-DNA insertion in an intron. Thus, the authors must complement the root phenotypic analysis with a qRT-PCR in order to assess if the mutant lines actually show a decrease in the gene’s transcript when compared to control plants.

Response: Isolation and validation of the root phenotype of *yuc3* with a second homozygous allele was not possible because of limited time for revision. Therefore, we decided to remove the *yuc3* dataset from the manuscript. This change does not affect any of the major messages of our manuscript.

- In Supplementary Fig.3, the authors observed that *yuc3*, *yuc5*, and *yuc7* mutants show a decrease in primary root and LR lengths under N deficiency. However, the *yuc-5* mutant has a Col-3 background and the authors compare it with Col-0. Thus, the authors must compare the response of the *yuc-5* mutant line with a Col-3 line.

Response: Thanks for spotting this. We repeated the experiment by comparing the root responses of *yuc5-1* (SAIL_116_C01) to mild low N with Col-3. We found that root elongation including primary root and lateral roots of *yuc5-1* was substantially decreased compared with wild type under low N. These results are now included in the Supplementary Fig. 3 and mentioned in the revised text (*Lines 119-121*).

- Supplementary Table 1 with information about the accessions used in this study is missing and Supplementary Table 2 containing information about SNPs in YUC8 coding sequence are missing. Supplementary Table 3 with information about the two major haplotypes is also lacking. Supplementary Table 4 with the primers used in this study is missing as well.

Response: From our perspective, it appears that there was a problem only for Reviewer 2 to access these Supp. Tables. We are sorry for this but confirm that the tables were and are still present.

- Line 149 is not finished. The position of the substitution must be stated.

Response: We apologize for this mistake. We now added the “position 14” in the revised version (*Line 189*).

- The authors should report the experiments by which they confirmed *yucQ* transformation with the YUC8-hap A and YUC8-hap B haplotypes.

Response: As stated in *lines 206-210*, we assessed the expression of *YUC8* in several independent transformants by *qRT-PCR* and selected for the phenotypical analysis three independent T3 homozygous lines showing comparative expression for each construct. The results were presented in the Supplementary Fig. 12a of the previous and Supplementary Fig. 18a in the revised version of manuscript.

- The authors transformed *yucQ* mutants with a Col-0 promoter in Figure 3. However, they did not specify the *yucQ* background. If this mutant did not have a Col-0 background, how can the authors be sure that phenotypes observed in *yucQ* and *yucQ* complemented lines are comparable with Col-0? The *yucQ* background needs to be specified.

Response: The comparison of *yucQ* with Col-0 has been widely used in previous studies and therefore appeared to us as being accepted by the community (see e.g., Matosevich et al., Nat. Plants, 2020; Gaillochet et al., 2020, Development; Zhu et al., 2019, J Biol. Chem.; Paris et al., 2018, Front. Plant Sci.; Tsugafune et al., 2017, Sci. Report; Liu et al., 2016, Plos Genet.; Sugawara et al., 2015, Plant Cell Physiol.; Li et al., 2012, Genes & Dev.). Nonetheless, we tracked back the genetic background of the single mutants and found that it is Col-0 for *YUC3*, *YUC7*, *YUC9* and most probably for *YUC8* (assigned only as Col), while it is Landsberg for *YUC5*. Unfortunately, the original paper by Chen et al. (2012) does not provide information on the order of the crossings to estimate eventual background effects. However, we note that the main conclusions based on our *yucQ* experiments are not to show an additive effect of several YUCs to the root phenotype but rather to serve as a multiple *yuc* deletion line, in which we quantify the relative contribution of *YUC8* haplotypes under low N versus sufficient N. Building our conclusion on this relative assessment makes the impact of the genetic background less relevant, since it just determines the size of the biological effect. We now added this information in the revised manuscript, *lines 369-370*.

- In figure 3e-f, authors compared PR and LR length between mutant and complemented genotypes. The authors concluded that *YUC8*-hapA variant rescued more efficiently PR length. However, the numbers used in these experiments are not comparable. Despite agreeing with the mentioned conclusions, we suggest performing a statistical analysis with a comparable N number.

Response: To address the reviewer's concern, we now also performed a Welch's *t*-test to specifically compare the lines complemented with *YUC8*-hap A and *YUC8*-hap B (note that both had similar N numbers of 110 and 113, respectively). This additional statistical test provided further support that the *YUC8*-hapA variant is more efficient in rescuing PR and LR elongation of *yucQ*. The Figure 3e-f has been revised accordingly.

- In Figure 4 (a-c), the letters showing statistical differences are too big and the box plots too small, making it difficult to read the graphs. The same can be said for Figure 4 e-f; Uppercase letters should be placed next to the boxplot. The graphs in this Figure must be improved.

Response: Thanks. We now decreased the size of letters and reformatted the Fig. 4 in the revised manuscript.

The letters showing statistical differences in Fig. 5 b and c are too big when compared to the boxplots. The authors should improve the format of the graphs.

Response: Thanks. We now decreased the size of letters in the revised Fig. 5b-c.

- Most of the discussion is focused on explaining why the *YUC8*-Hap A is associated with LR growth under a low N concentration, and states the novelty of this study. However, the authors poorly contrast their results with the known signaling pathways, molecular components and already

proposed models of N signaling underlying root responses. Are there any N- responsive transcription factors that could be candidates for the control of BR - associated genes?

Response: As suggested by the reviewer, we now also refer more extensively to other signaling pathways in the revised text (*Lines 147-158* in Results where we present the results of *AGL21*; and *Lines 350-353* in Discussion where we refer to CEP-CEPRs-CEPDs systemic N signaling pathway). However, in order to accommodate further points of discussion requested by other reviewers and avoid simple comparisons between pathways that respond to different N scenarios, we did not expand too much the discussion on known signaling pathways that are not specifically involved in the regulation of root foraging responses to mild N deficiency.

Are these results consistent with any other models of auxin - mediated control of root architecture in response to N availability? How can the study's results be incorporated into these already proposed models?

Response: Thus far, most known auxin-mediated control of root development (e.g., *miR167/ARF8* and *miR393/AFB3* modules regulating LR initiation and progression; and *NRT1.1*-dependent auxin transport in LR emergence) are restricted to responses to high nitrate or to severe N deficiency (Gifford et al., 2008; Vidal et al., 2010; Krouk et al., 2010; Bouguyon et al., 2015). As mentioned above, we also show in the revised version that *AGL21* and *ANR1* do not play a significant role in root growth responses to mild N deficiency and *YUCCA*-dependent local auxin biosynthesis is not required for root survival response to extremely low N availability. Therefore, our results and the findings of previous study suggest that distinct signaling modules coordinate the characteristic changes induced by nitrate supply, or by moderate or severe N limitations. We now discuss these aspects more clearly in the revised manuscript (*Lines 147-158 and 278-281*).

- Line 66-67: Plants are not grown in high or low N for 9 days. Plants are grown for one week on sufficient N, plus 9 days on HN or LN. Please change to make it more clear to the reader.

Response: Agreed. We now amended the sentence to "After transferring 7-day-old seedlings precultured on sufficient N to HN or LN for 9 days" *Lines 79-80*.

- Fig 1c: This figure is not showing relative average LR length, as indicated in the legend. Please change "relative average LR length" for an x-axis title "LN/HN LR length ratio"

Response: Thanks, done!

- Line 82: "we nonetheless analyzed" I think this conjunction is not well used in this context. Maybe change to "we then analyzed..."

Response: Thanks, done!

- Supp Fig. 2ab: There are no boxes indicating what the colors mean.

Response: Thanks for spotting this mistake. We now include a legend on the top of the Supplementary Fig. 2a,b.

- Supp Fig. 2A should be included in Fig 1, since it is part of the root morphological analysis of the WT vs. *YUC8* mutants. Total root length graph is redundant since there is already a plot showing PR length and LR length.

Response: Considering that lateral root number is not significantly different between WT and *yuc8* mutants and that Fig. 1 is already quite crowded, we prefer to leave these results in Supplementary Fig. 2.

- Figure 2: You have already shown a LR phenotype for the *yuc8* mutants, why center the cell elongation analysis on a mutant that includes genes that are not expressed in roots? The *yucQ* mutant is associated with an important decrease of IAA content that can mask a more-specific, YUC8-N-related phenotype.

Response: In Fig. 2a-d, our intention is to estimate the role of YUCCA-dependent auxin biosynthesis in increasing meristem size and cortical cell length to stimulate lateral root elongation.

- Line 112: "Conversely, when YUCCA-dependent auxin biosynthesis in roots of wild-type plants was suppressed with 4-phenoxyphenylboronic acid (PPBo)..." Since PPBo is an inhibitor of YUCCA activity, how can you be sure that auxin biosynthesis is specifically suppressed in the roots?

Response: As shown in Supplementary Fig. 7, to inhibit auxin biosynthesis in roots, we supplied PPBo to the root-containing agar while removing the agar on the top part containing the shoots.

- Supp Fig. 8. What happens with PR length response to low N? Is it also altered by NPA (I can't really tell from the photograph)? A graph of PR root length should also be added to Supp Fig 8 to support the conclusion that "these results collectively indicate that YUCCA-mediated local auxin production in roots modulates root elongation under mild N deficiency"

Response: Compared to mock, the supply of 5 μ M NPA to the shoot compartments slightly but significantly inhibits PR elongation under both low and high N. However, the response of PR length to low N (i.e., LN-to-HN ratio) was not altered by NPA supply to shoots, corroborating that polar auxin transport did not affect low N-induced root elongation. Following the reviewer's recommendation, we now include this dataset in revised Supplementary Fig. 10.

- This paragraph (l. 128-137) should be included before the analysis of gene expression, to support the conclusion that YUCCA-mediated local auxin production modulates root elongation under mild N deficiency (l.118-119).

Response. Thanks for this suggestion. However, we think that the increased root auxin accumulation revealed by the *R2D2* reporter is a consequence of enhanced expression of *YUC* genes under mild N deficiency. Following this logic, we prefer not to modify this paragraph.

- Supp Fig. 9: Please include panel a) on Fig 2e

Response: Thanks. GUS staining was performed at both 7 and 9 days after transfer to N treatments. To remain consistent with the LRs shown in Fig. 2e in previous and Fig. 2i in present version of manuscript, we moved stained PRs from plants grown for 9 days on the treatments to Fig. 2i.

Reviewer #3 (Remarks to the Author):

In this manuscript, Jia et al. takes advantage of different *Arabidopsis* accessions carrying natural variation to evaluate their root response to nitrogen deficiency, intending to seek the components involved in this biological process. Using GWAS and SNP analysis, the authors identified a target gene YUC8, an auxin biosynthesis-related gene, and proposed that this gene's family members contribute to lateral root elongation response to N deficiency. The genetic experiments using *yuc* single or multi-mutants further confirmed that the YUC-mediated local auxin biosynthesis process in root is associated with the root response to nitrogen. Consistently, the YUC genes' expression levels could be triggered by N deficiency. Meanwhile, the authors identified two allelic variants of YUC8 harboring a non-synonymous substitution of one amino acid. They demonstrated that the different biological activity between YUC8 variants underlies the differential lateral root foraging for nitrogen for these *Arabidopsis* accessions. Moreover, the genetic combined with the pharmacological experiments imply the auxin signaling pathway is downstream of the BR signaling in the root foraging for nitrogen. Generally, this work seems interesting, especially the crosstalk signaling between plant hormones

brassinosteroid and auxin in root foraging for N. The data are also well organized and the paper is very easy to be followed.

Response: Thank you for the positive comments on our work.

However, some of the conclusions drawn from the data are not so persuasive, and I suggest the authors revise the text or add much more data to make it more explicit and accurate. Meanwhile, some analyses are a bit superficial. I have some comments below for authors to improve.

Major comment

1. The information conveyed by this title is not exactly what the data can support. The GWAS analysis results only conclude that the allelic variants of *YUC8* contribute to the differential local auxin biosynthesis activities among these *Arabidopsis* accessions and thus trigger their different root foraging for nitrogen, but it cannot be used as the regulation mechanism in plants as title proposed. Concerning the mechanism, from the existing data, I believe that the auxin biosynthesis triggers root foraging for nitrogen, but no data here shows that only *YUC* genes are up-regulated in the mild N deficiency. Are the expression levels of other crucial auxin biosynthesis-related genes (e.g. *TAA*) up-regulated under nitrogen deficiency? Do their corresponding chemical inhibitors (e.g. L-kyn inhibits *TAA* activity) also disturb the lateral root elongation in low nitrogen as *YUC* inhibitor PPBo shows? If yes, it should be very careful to use the “*YUC*-dependent local auxin biosynthesis”, because other auxin synthesis genes are also involved in this process. Therefore, more experiments should be considered to enhance the view as the title proposed or modify the title.

Response: Thank you for these constructive suggestions. According to Ma et al. (2013, Plant J), both *TAA1* and *TAR2* are induced by mild N deficiency in the roots. In the same study it was also shown that *TAR2* is responsible for LR emergence but does not affect N-dependent LR elongation, probably because expression of *TAR2* is restricted to the root maturation zone. Instead, *TAA1* is expressed in the vasculature and root apical meristem, which overlaps with expression domain of *YUC8* and its homologous genes (Yang et al., 2014, Plant Cell; Chen et al. 2014, Plant Cell Physiol.; Brumos et al., 2019, Dev. Cell; Brumos et al., 2020, Plant Cell). In the revised version, we confirmed that *TAA1* is up-regulated by mild N deficiency and demonstrate that it is also under the control of the BR signaling transcription factor *BZR1* (new Supplementary Fig. 8). Furthermore, we assessed the root phenotypes of two *TAA1* mutant alleles (*ckrc1-1* and *wei8-1*) and found that they are also unable to stimulate root elongation in response to mild N (new Supplementary Fig. 9), suggesting a major role for *TAA1* together with *YUC5/7/8* in this process. Taken together, these results suggested that local auxin biosynthesis produced by *TAA1*-*YUC5/7/8* module acts downstream of BR signaling in stimulating root elongation in response to mild low N. We consequently amended our title to “Local auxin biosynthesis acts downstream of brassinosteroids to trigger root foraging for nitrogen”.

2. Usually, the genetic variations of geographically representative *Arabidopsis* accessions confer their fitness in a certain habitat. However, I did not see any this kind of analysis throughout the paper. To go much deeper, I urge the authors to further check the geographical distribution between the two groups of *Arabidopsis* accessions carrying the *YUC8* variant *YUC8*-hapA and *YUC8*-hapB separately. Then combined with the feature of the local environments, it might provide some insights regarding how the auxin-dependent root foraging for N is modified by natural selection and linked to their adaptability to local environments. At least, it should be discussed in the discussion section.

Response: Thanks for this interesting suggestion. We exploited data of a previous climate adaptation study (Hancock et al., 2011) and observed that there were indeed significant positive correlations between the SNP_Chr4_14192732 and temperature seasonality and minimum temperature (new Supplementary Fig. 24a,b). Furthermore, we detected a clear association between the distribution of *YUC8* allelic variants and temperature-related variables. As we now show in new Supplementary Fig.

24c-f, accessions harboring the YUC8-L allele grow on habitats with higher mean diurnal range, temperature seasonality, temperature annual range and mean temperature of wettest quarter as compared to accessions with the YUC8-S variant. These observations may suggest that YUC8 variants play an adaptive role under prevailing temperature variability and support the role of YUC8-mediated auxin biosynthesis in plant adaptation to temperature variation (Sun et al., 2014, Plos Genet.; Lee et al., 2014, Nat. Commun.; Fiorucci et al., 2019, New Phytol.; Bellstaedt et al., 2019, Plant Physiol.). These findings are now presented and discussed in *Lines 300-312*.

3. Performing experimental analysis with only Arabidopsis accession Col-0 to verify the GWAS results is not so convincing. If some other Arabidopsis accessions (e.g. Co) can be used for similar experimental analysis, the conclusions would be much more solid. For example, even though the bioinformatic analysis (GWAS, SNP) proposed that YUC8 coding variants (hapA and hapB) link to the diverse root response to N among these Arabidopsis accessions. However, no further data demonstrate if the differential transcriptional level of YUC8 is also involved in the diverse nitrogen responses or not.

Response: In order to assess the possible involvement of differential transcript levels of YUC8 in root responses to low N, we determined transcript levels of YUC8 in 9 natural accessions showing quantitative variation in their lateral root lengths and N responses. Of these 9 accessions, Co, Ty-0, Edi-0 and Tamm-27 express the YUC8-hap B variant, while Col-0, Ven-1, Kas-2, JEA and Uod-1 express the YUC8-hap A variant. According to our results i) there was no significant correlation between lateral root length and transcript levels of YUC8 at either N condition; ii) no significant correlation was detected between fold-change (LN-to-HN ratio) of YUC8 expression and lateral root length ratio; and iii) we found no difference in YUC8 expression between two YUC8 haplogroups at either N condition. We presented these results in Supplementary Fig. 10 in the previous manuscript and Supplementary Fig. 15 in the revised manuscript. Together, these results suggest that in the investigated lines YUC8-dependent natural variation under LN is unlikely due to variations at the transcript level.

Even though to request some transgenic work is not so reasonable, some other simple experiments can be performed using other Arabidopsis accessions. Is the transcription level of YUC8-hapB in accession Co also up-regulated under low N condition? Is its increased expression level of YUC8-hapB comparable to that of YUC8-hapA in Col background?

Response: i) We found that YUC8 transcript levels in the accession Co increased by 17%, which is not significantly different between the two N conditions. ii) In Col-0, YUC8 increased by 41%, which was not much different to Co. Furthermore, we selected lines with comparable expression for the allelic complementation experiment (Supplementary Fig. 18).

Moreover, to further prove that the different auxin accumulations of these Arabidopsis accessions lead to their varied root response to nitrogen, the exogenous 50nM IAA can be applied in different Arabidopsis accessions, to see if the elimination of endogenous auxin accumulation differences would abolish the differences of root N response among these Arabidopsis accessions.

Response: Thanks for this suggestion. Although external application of auxin to YUC8-hap A and -hap B accessions is expected to abolish the differences in endogenous auxin accumulation, the interpretation of the results will be complicated by genotypic differences in auxin uptake, transport and signaling. Therefore, we decided to supply the YUCCA inhibitor PPBo (5 μ M) to roots of 8 accessions from YUC8-hap A (Uod-1, Or-0, Wt-5, Kas-2, JEA, Ven-1, Pog-0, Ri-0) and 6 accessions from YUC8-hap B (Co, Edi-0, Tha-1, Ty-0, Alst-1, Tamm-27). Whereas under mock treatment the root foraging responses (i.e. LN-to-HN ratio of PR, LR and total root length) of YUC8-hap A accessions were on average significantly stronger than those of hap B accessions, exogenous supply of PPBo

strongly abolished these differences. These results are now presented in the new Supplementary Fig. 20 and mentioned in *Lines 217-223*.

4. Something needs to be further clarified. The author assessed 200 Arabidopsis accessions and then identify YUC8 is involved in the diverse LR response to low N among these accessions. However, to identify the SNPs within YUC8 coding sequence, they used 139 Arabidopsis accessions. It is better to explain why different numbers of accessions are used for these two analyses and what criteria they applied for selecting the accessions to perform these bioinformatic analyses.

Response: We apologize for not clearly explaining this point in the first version. We did not apply any criterion to select 139 accessions for YUC8-based association analysis. In the GWAS analysis, we phenotyped 200 accessions that are genotyped with a 250k chip. However, of these 200 lines, complete genomic information is available for only 139 accessions, which have been re-sequenced (1001 Genomes Project; <http://signal.salk.edu/atg1001/3.0/gebrowser.php>). This point is now mentioned in the Methods. *Lines 411-414*.

Minor comment

1) Some descriptions are a bit overstated in the Abstract section. Page 1 line 11, it should be the “flowering plants”, not the “plants”.

Response: Thanks, we changed to “flowering plants”.

Line 32, this work only implies the ‘root response to nitrogen availability’ but not the ‘root response to abiotic stress’.

Response: Thanks, we changed to “nitrogen availability”.

2) The concentration of these chemicals used for plant treatment should be clearly showed. It is better to mention them when they appeared in the main text for the first time. For example, it is very hard for me to find which concentration of NPA they used in both the main text and figure legends.

Response: Thanks for pointing this out. We now specified this information in the text and figure legends.

3) Line 229, it is hard to have the conclusion that “mild N deficiency stimulated local auxin biosynthesis in the root apical meristem”. Firstly, no evidence shows that the auxin biosynthesis genes (e.g. YUC) are only up-regulated in root apical meristem. Secondly, even though the R2D2 signal is enhanced in RAM, however, it is also possible that the excessive synthesized auxin in other tissue is transported to RAM. It is much accurate to say the “local auxin accumulation in the root meristem”

Response: Thank you for this suggestion. We now say “local auxin accumulation”.

4) Line 274, there are few minor grammar mistakes, check through the manuscript to correct them. e.g. Line 274, ‘were’ should be changed to ‘was’. Line 149, some words seem to be missing after ‘at’.

Response: Thanks. Done!

5) Some words are very hard to be digested. ‘Homozygosity’ should be replaced by ‘homozygote’.

Response: Thanks. Done!

Reviewer #4 (Remarks to the Author):

The manuscript of Jai et al describes the natural genetic variation in YUC8 is involved in auxin

biosynthesis as cause for the root foraging response to low N. Overall, manuscript is well written, authors have done a great job by doing many detailed and in-depth experiments to support their findings and conclusions.

Response: Thanks for the encouraging comments about our work.

However, I have a number of questions/comments regarding the data presented and there are still some issues that authors should take into account.

1) Authors only focused on one GWA peak located on chromosome 4 where top marker SNP located close to YUCCA8 gene. However, GWA analysis clearly also identified other regions with significant association (on Chr. 4 and Chr. 2).

Response: The SNP on the Chr2 is not significant when a False Discovery Rate at $q = 0.05$ is applied. As such, we detected two significant SNPs on Chr 4 (SNP_2724898 and SNP_14192732). Among these, we focused on the SNP Chr4_14192732, as the corresponding peak was supported by adjacent markers and T-DNA insertion lines were available for all genes falling within a 20-kb supporting interval. We clarified this point in the revised text. *Lines 86-91.*

Even though further experiments carried out meticulously by authors to show YUC8 as determinant for LR response to low N. In order rule out bias approach to choose YUC8 region for follow up experiments, authors should also investigate whether other genes located close to top SNPs from other two significant associations (shown in Fig. 1d.) are contributing to the LR response to LN or not. Authors need to perform expression analysis under HN/LN and phenotyping using T-DNA insertion lines of the genes falls within 10 to 20 kb regions from top SNPs present on Chr. 2 and Chr. 4.

Response: Indeed, there were two further genes (At4g28730 and At4g28740) falling within a 20-kb supporting interval of the investigated SNP_Chr4_14192732. We investigated T-DNA insertion lines of both genes and observed that LN-induced LR length was similar to wild-type plants, and the expression of either gene did not respond to LN (Supplementary Fig. 1b-e), excluding an eventual role of At4g28730 and At4g28740 in regulating LR elongation under mild N deficiency. We did not consider genes falling within a similar interval of the other SNPs, because the outcome of our experiments revealed a significant impact of YUC8. Thus, our discovery of relevant YUC8 variation and its contribution downstream of BRs remains valid irrespective of a putative contribution of genes in another locus. We believe that unravelling the contribution of the other SNP is matter for another story and beyond the scope of this manuscript. To clarify our decision to focus on SNP_Chr4_14192732, we improved the description in *Lines 89-102.*

2) Results section is very well written but introduction and discussion sections are too short. Authors should rewrite both the sections. Authors can further enrich the introduction part by giving better background on the LR development and N availability. For example: adding “role of TAR2 in low nitrogen-mediated reprogramming of root architecture in Arabidopsis” etc etc.

Response: Thanks for this suggestion. We now substantially expanded the Introduction and Discussion.

3) Authors should also discuss Auxin-BR crosstalk in more detail instead of writing just few lines on this interesting aspect. Also, in this case authors can prepare a schematic illustration describing the crosstalk between auxin-BR signaling and its role in LR growth under LN.

Response: Thanks for this suggestion. We revisited and substantially revised this aspect and now provide a schematic illustration in the revised manuscript (new Figure 6).

4) Figure 4 seems very complex, I would strongly suggest authors to replot this figure in a different way for better understanding for the readers.

Response: Thanks. We now reformatted Fig. 4 in the revised manuscript.

5) In Methodology section sufficient information is not given, for example haplotype analysis part is too short.

Response: We apologize for not being sufficiently clear. We performed *YUC8* gene-based association analysis by using identified SNPs from 139 natural accessions whose *YUC8* sequence is available to download from the 1001 Genomes Project (<http://signal.salk.edu/atg1001/3.0/gebrowser.php>) and have phenotypic data in our initial screening. The 6 resulting significant SNPs were taken to classify *YUC8* haplogroups. We only considered those haplotypes, for which more than five accessions were available for further comparative analysis. This information has now been included in the revised Methods. *Lines 411-420.*

Also, in many experiments sample size varies a lot and in some cases number of individuals are not optimum in my opinion. It is important to mention in that case whether experiments have been repeated multiple times or not.

Response: In Fig. 3e-g, Supplementary Fig. 11 and Supplementary Fig. 12 c-e in the previous version (Fig. 3e-g, Supplementary Fig. 17 and Supplementary Fig. 18 c-e of revised version), we pooled all 6, 20 and 3 independent transgenic lines expressing the same construct. Following the reviewer's recommendation, we now clarified these points in the corresponding legends. Furthermore, we repeated all experiments at least once and presented the results from one representative experiment.

6) Authors last paragraph (Line 267-270) in discussion section is superficial and overclaiming.

Response: We deleted this paragraph and added instead the following sentence: "Under any of these constraints, employing CRISPR-mediated gene editing to turn "weak" *YUC8* variants into "strong" variants could provide an opportunity to increase root elongation and subsequent water and nutrient acquisition in crops". *Lines 355-358.*

REVIEWERS' COMMENTS

Reviewer #1 (Remarks to the Author):

I would like to say well done to the authors in pursuing the improvements of their manuscript and am glad that my comments were useful. In this revision the authors explain the rationale for the choice of the 200 accessions and discussion of hap A vs. hap B accessions; I am sure that this will continue to be an area of interest as we learn more about more subtle molecular differences amongst accessions in the future. In addition they help better place the work in context of previous physiological-focussed BR-auxin-N interactions, addressing the major comment of reviewer #2, explained the choice of significant SNP region (all reviewers), further elaborated on the evaluation of YUC8 expression levels (reviewer #3) and developed the detail in the text (reviewer #4). All of these have improved the manuscript and I enjoyed the discussion provided in the response to reviewers.

I am sure that the manuscript will be of broad interest to both plant scientists and any researchers interested in how molecular signalling is integrating at the local vs. systemic level to control a phenotypic response.

Reviewer #2 (Remarks to the Author):

Thanks to the author for constructively addressing my comments. This version of the manuscript satisfies my concerns.

Reviewer #3 (Remarks to the Author):

The author(s) have addressed all my concerns, and generally, the revised manuscript is greatly improved. In this version, the author(s) analyzed the other auxin biosynthesis genes and indicated that the local auxin biosynthesis is involved in the root foraging for nitrogen. They further evaluated the geographic distribution of the two groups of Arabidopsis accessions separately carrying YUC8-hapA and YUC8-hapB, suggesting that the genetic variations link to the plant adaptability to temperature differences. The authors also give detailed information regarding the Arabidopsis accessions used for the phenotypic and GWAS analysis. Meanwhile, the conclusion is much more solid and the interpretation is more precise after the revision.

I only have some very minor points irrelevant to the academic question, which should be considered by the author for improvement.

1. In Abstract (Line 19), please give the full name of the 'TAA'.
2. In Supplementary Fig 8 and Supplementary Fig. 23, for these statistical analysis, the symbols 'ns', '*' and '**' above the bars confused me a lot, because they were everywhere. I am not so sure which data sets were used for comparison to see the significant differences. Author should clarify it in relevant figure legends to make it clear.

Reviewer #4 (Remarks to the Author):

It appears to me that most of my concerns have been appropriately addressed by authors and the manuscript has been greatly improved.

REVIEWER COMMENTS

Reviewer #1 (Remarks to the Author):

Scope and novelty of research:

Jia et al investigate the foraging response in roots; this is a long-known phenotype that enables roots to preferentially grow in high(er) areas of nitrogen (or other nutrients) in order to overcome the lack of nitrogen in other areas of the root system. The authors find that the enzyme-encoding gene YUCCA8 regulates auxin biosynthesis in lateral root tips, which intersects with a brassinosteroid (BR) signalling module. Crosstalk between these hormonal networks helps control the foraging response and this operates differently in lateral roots compared to the primary root. The authors implicate particular BR signalling components in this intersection. This work is within scope for Nature Communications since it uncovers both new genes and interactions, but also does this by investigating natural variation to identify a new communication hub. This work will help others to approach investigating the foraging response in a new way. It should be of wide interest, not only to plant biologists but to those studying natural variation and those interested in hormonal signaling in any multicellular organism. Although the discovery that auxin-production/gradients in roots is required for responses to the environment and that the BR and auxin pathways intersect are not novel per se, the discovery that this response is modulated by differences in coding region (rather than expression level or expression regulation) is novel and very interesting. In addition, the *bsk3/yuk3* mutant analysis was an elegant way to assess intersection between these pathways, following up with *bsk* analysis to pinpoint the location of effect to the lateral roots specifically. This makes the findings and their implications novel. The huge amount of work included was all well-carried out, methods well-compiled, and I found the conclusions to be well-supported. All numerical data was well-displayed and catalogued, with appropriate tests being carried out in order to assess significance. Indeed, the use of a randomised block design for plant growth was a high level of control that is not typically seen.

Response: Thanks for the encouraging comments about our study.

The putative ER-localisation is interesting as the ER is being increasingly found to act as a hub to shape functions of the cell via affecting responses to the environment and how proteins are stored or mobilised. I expect future work could be done to investigate this at a cell biology level, using tagged YUC8-hap variants.

Response: Thanks for this interesting suggestion. Indeed, we are planning to continue research on the identified YUC8 variants, which will include intracellular localization studies.

Major comments:

The only aspect I found to be less well-developed was description of why the 200 accessions were chosen – it was stated that these were chosen as they were diverse (line 66) but what they were diverse in was not described; diversity at the sequence level?

Response: We apologize for not being clear. In our institute, a genetic diverse collection of 200 *Arabidopsis thaliana* accessions has been assembled, which is based on geographical distribution (as described in Jia et al., 2019, Nature Commun.) and which have been genotyped with a 250 k chip. To clarify this point in the manuscript, we rephrased the text to “in a geographically and genetic diverse panel of 200 *A. thaliana* accessions...” (Line 77).

Also, at the end, I wanted to know more about the hap A vs. hap B accessions – do these differ in any other phenotypes (from what the authors know)?

Response: While scoring lateral root length for these 200 lines, we also assessed other root traits such as primary root length, total lateral root length, total root length and lateral root number under HN and LN conditions. In addition to longer average lateral root length, YUC8-Hap A accessions had on average also longer total root length and total lateral root length (which are related parameters) under low N than YUC8-Hap B accessions. However, there was no significant difference for primary root length and lateral root number. There was also no difference in the shoot phenotype or in shoot biomass. We now included this piece of information in Supplementary Fig. 16 and in the revised text (*Lines 196-197*).

I could not see anything (by eye) when looking at latitude/longitude data for these, but what about their responses in other known/published studies of root/foraging/nitrogen responses?

Response: To address this question, we checked the response of the contrasting lines identified here in other GWAS reports presenting phenotypical data for root traits (Gifford et al., 2013, Plos Genet., Satbhai et al., 2017, Nat. Commun.; Li et al., 2019, Nat. Commun.; Ogura et al., 2019, Cell). However, with regard to root morphological responses to Fe deficiency/toxicity, nitrate starvation or auxin transport inhibition by NPA, we did not find substantial genotypic differences for our contrasting lines.

Minor comments/typographical

Line 84: deletion of YUC8 – this should be loss of expression of YUC8, not deletion

Response: Thanks. We now rephrased the text to “loss of *YUC8* expression” (*Line 102*) in the revised manuscript.

What is the other significant SNP on Fig.1d – it is not a peak so probably irrelevant, but perhaps worth noting in the legend.

Response: Actually, we did not consider it further as the peak consisted only of one SNP. But we will check it again. We now noted in the revised manuscript that we chose the peak with further supporting SNPs (*Lines 86-91*).

Fig.1d – it would help to label At4g28720 as YUC8, and just use black text rather than the teal colour (or it looks like these should be on chromosome 2).

Response: Thanks for pointing to this. We followed this suggestion and revised Fig. 1d.

Figures: What are the black dots/small dashes on the plant seedling images? They are on more, although not all images. They are generally consistent within sets of data, excepting in Supp Fig.7.

Response: The black dashes/dots represent the position of the primary root tip at the time when the seedlings were transferred from the pre-culture to the N treatments. We now added this information in the Material and Methods (*Lines 387-389*).

Line 201: ‘they were not anymore upregulated by LN’ – change to ‘they were not upregulated by LN as they would be in WT plants’ (since ‘anymore’ could suggest any more, meaning an expression level effect, rather than a yes/no effect)

Response: Thanks, done!

For some of the supplemental data values I would display the data at the limit of measurable resolution e.g. 2 decimal places for PR values (e.g. Supp Fig.11 data) and 3 decimal places for LR

values (e.g. Fig.1 data, Supp Table 3 or Supp Fig.15 data). This does not affect or change any of the figures themselves, or even the data included, just the format it is displayed in.

Response: Thanks, done!

Reviewer #2 (Remarks to the Author):

Review for Nature Communications

YUCCA-dependent local auxin biosynthesis downstream of brassinosteroids triggers root foraging for nitrogen

Zhongtao Jia, Ricardo F.H. Giehl, Nicolaus von Wirén

Summary

Nitrogen (N) is an essential macronutrient that modulates many plant developmental processes. The development of lateral roots (LR) is strongly determined by nitrogen availability. Plants grown under N deprivation develop longer LRs to reach nitrogen-rich patches. Mechanisms for nitrogen-mediated growth of LR have been reported before. However, there are still many details that remain unclear. In this manuscript, Jia et al. explored how YUCCA8 (YUC8) modulates LR elongation in response to N deficiency. The authors show that N-dependent auxin biosynthesis in LRs acts epistatic and downstream to a canonical brassinosteroid signaling cascade. Even though the study shows that the methods used are appropriate to identify genes related to signaling pathways, the connection between YUCCA, auxin and N has already been reported.

Response: Indeed, previous studies have reported that auxin levels increase in roots of low N - exposed plants (Caba et al., 2000, *Planta*; Walch-Liu et al., 2006, *Ann. Bot*; Tian et al., 2008, *J. Plant Physiol*; Kiba et al., 2011, *J. Exp. Bot.*). In part, these studies even suggested that the higher auxin levels detected in N-deficient roots originate from increased shoot-to-root auxin transport (Forde, et al., 2002; *J. Exp. Bot.*; Kiba et al., 2011, *J. Exp. Bot.*). However, all these observations remained at the physiological level. One exception is the study of Ma et al. (2013, *Plant J*) showing that TAR2-dependent local auxin biosynthesis is induced by mild N deficiency. However, in *tar2* mutant plants lateral root emergence but not lateral root elongation is impaired. To the best of our knowledge, our findings that YUCCA proteins stimulate local auxin biosynthesis in response to mild N deficiency and that this activity has a specific effect on lateral root elongation is completely novel (see additional comments below).

Even though the interplay between BR and auxins in the root response to mild N deprivation is new, the results do not present a major breakthrough in the understanding of the molecular machinery that orchestrates the interplay between N and phytohormones in the control of root system architecture.

Response: We disagree with the reviewer's view. The synergistic action of BRs and auxin in lateral root elongation was not documented before. Regarding this N deficiency response, we could even place auxin biosynthesis downstream of BR signaling, which is not the case for most other studies having reported on BR-auxin crosstalks that primarily involves auxin signaling and transport (Nakamura et al., 2003, *Plant Physiol.*; Bao et al., 2004, *Plant Physiol.*; Nemhauser et al., 2004, *Plos Biol.*; Mouchel et al., 2006, *Nature*; Vert et al., 2008, *PNAS*; Zhou et al., 2013, *Mol. Plant*; Cho et al., 2013, *Nat. Cell Biol.*; Retzer et al., 2019, *Nat. Commun.*). Moreover, our study demonstrates that YUC-dependent auxin biosynthesis represents a novel key element in this signaling cascade, which

together with our previous findings (Jia et al., 2019, Nat. Commun.; Jia et al., 2020, Plant Physiol.) now shows that *in-situ* biosynthesis of BR and auxin determine N-dependent root elongation. This advances our understanding relative to previous studies, which proposed that polar auxin transport was key for N-dependent changes in root morphology (Forde, et al., 2002, J. Exp. Bot.; Kiba et al., 2011, J. Exp. Bot.). Beyond that, our study shows the existence of natural variation for YUC8-driven auxin biosynthesis which greatly impacts on the adaptation of root traits to mild low N. The uncovered N-dependent, YUC-mediated local auxin biosynthesis in roots downstream of BR signaling represents a novel regulatory module and thus a significant advance in the mechanistic understanding of root plasticity changes in response to environmental cues.

Major comments

- The authors do a great job characterizing the root response to mild N availability and the role of YUCCA genes. However, the crosstalk between nitrogen nutrients and different phytohormones in the control of root architecture has been described before. For instance, Yu et al. showed in 2014 that YUCCA and N deprivation are downstream of AGL21. The authors do not refer in detail to the known aspects about N-dependent control of root architecture in both the introduction or discussion. Therefore, these sections seem to be out of context dismissing prior work by several different groups.

Response: We regret that we did not detail all prior work. Indeed, we agree with the reviewer that the study of Yu et al. (2014) is relevant for our story. Therefore, we now included it in the revised Introduction (*Lines 53-54*) and Results (*Lines 147-158*). In addition, we have extensively revised the Introduction and Discussion sections to address the reviewer's concerns.

- The connection between YUCCAs and mild nitrogen deficiency is undeniable in this work. Authors confirmed a previously reported experiment that established a possible interplay between YUCCA, nitrogen and auxin.

Response: We disagree with the reviewer that the connection between YUCCA-dependent auxin biosynthesis and mild N deficiency is just a confirmatory finding. To the best of our knowledge, previous studies did not show if and which YUCCAs are responsive to N deficiency nor did they address which YUCCAs are critical for root architectural modifications induced by low N availability. Besides addressing these aspects with independent approaches, we also identified natural YUC8 haplotypes that drive differential N-dependent lateral root elongation at the species level. Therefore, we believe that our results extend our knowledge well beyond what was known before.

Maybe, the reviewer's comment refers to the study of Yu et al. (2014, Mol. Plant), which showed that the transcription factor AGL21 regulates the expression of some YUCCA genes and root auxin accumulation, and demonstrate that AGL21 modulates lateral root growth under a N-free condition. However, this study did not investigate if the expression of YUCCA genes is induced by N starvation nor if *yucca* mutants show altered lateral root growth in response to N deprivation. Furthermore, the phenotypes of *agl21* mutant and AGL21 overexpressing lines were recorded under N-free, i.e. severe N deficiency, which induces distinct root architectural changes compared to the mild N deficiency that was investigated here. However, we took up the reviewer's comment and investigated whether AGL21 and its paralogous gene ANR1 are required to stimulate root elongation in response to mild N deficiency by using the *agl21 anr1* double mutant described previously (Liu et al., 2020, J. Exp. Bot.). As shown in new Supplementary Fig. 11, we observed that the *agl21 anr1* double mutant showed a similar response to mild N deficiency as wild-type plants. Furthermore, we phenotyped the *yuc8*

mutants and the *yucQ* mutant under severe N deficiency, a condition similar to that used by Yu et al. (2014, Mol. Plant). The length of primary root and lateral roots of all tested mutants was not significantly different to wild type (new Supplementary Fig. 12 and 13). Taken together, these results show that distinct signaling pathways are involved in root adaptive responses to mild and severe N limitations. These new results and the reference to the study of Yu et al. (2014) are now presented in *Lines 147-158*.

Thus, the most novel conclusions defined by the authors is the interplay between BR, auxin, YUCCA and N deprivation (Figure 5). However, while data presented supports a correlation there is not enough evidence to understand cause-consequence connections between these pathways. This reviewer believes additional experiments are required to demonstrate and understand the connection between YUCA and BR in LR length, which is the main novel aspect of this work. There are no insights about how N local perception and signaling is taking control of BR response to regulate LR growth. Is N regulating the expression of any genes associated with BR biosynthesis or signaling in roots in these experimental conditions?

Response: As mentioned in the Introduction, our previous studies have shown that mild N deficiency systemically up-regulates the expression of critical BR biosynthesis genes, including *DWF1*, *CPD*, *DWF4* and *BR6OX2*, and induces the expression of the BR co-receptor *BAK1* (Jia et al., 2019, Nat. Commun.; Jia et al., 2020, Plant Physiol.).

Is this a direct or indirect regulation?

Response: We expect that mild N deficiency-induced biosynthesis of BR via *DWF1*, *CPD*, *DWF4* and *BR6OX2* is indirect, most likely mediated by an N-responsive transcription factor, but we don't know yet which one. Identification of such transcription factor(s) is beyond the scope of the present study.

Can N induce BR accumulation endogenously in roots?

Response: Considering the difficulty to detect BRs in roots from agar-grown *Arabidopsis* plants, in our previous study (Jia et al., 2020, Plant Physiol.), we assessed the transcript levels of *DWF1*, *CPD*, *DWF4* and *BR6OX2* commonly used as molecular marker of BR levels (Shimada, et al., 2003, Plant Physiol.; Kim et al., 2006, Plant Physiol.; Chung et al., 2011, Plant J; Poppenberger et al., 2013, EMBO J; Singh et al., 2014, Plant Physiol.; Martins et al., 2017, Nat. Commun.; Wei et al., 2017, Plant Physiol.). As these genes were up-regulated by mild N deficiency (Jia et al., 2020, Plant Physiol.), it is expected that endogenous BR accumulation in roots is indeed increased in response to mild N deficiency.

Is there any component of N - signaling pathway required for the action of the BR - auxin module?

Response: Given that the various components of the uncovered local BR-auxin module in roots are controlled by the plant's shoot N status, we expect that some of the known components of systemic N signaling are indeed involved in this regulation. For instance, lateral root elongation is stimulated in *cepr1* and *cepr1/2* mutants (Tabaka et al., 2014, Science), suggesting that systemic N signaling via the CEP-CEPRs-CEPDs cascade may be involved. We now discuss this point in the revised manuscript (*Lines 350-353*).

Specific comments

- Genetic backgrounds are relevant in this study but not clearly described. This information is essential to interpret results.

Response: Thanks for pointing this out. We carefully checked our mutant lines and found all mutants were in the background of Col-0, except for *yuc5-1* (SAIL_116_C01) which is in the background of Col-3. We clarified these details in the Methods (*Line 361*) and in the figure legends as well.

The authors should also clarify if their mutants are knockdown or true knockout lines. It is recommended to add a Supplementary Table to organize all this information including all mutants.

Response: Thanks for this suggestion. The *yuc8-1* (SALK_096110C) and *dwf4-44* (SAIL_882_F07) have been characterized and documented as knockdown mutant in Sun et al. (2012, Plos Genet.) and in Du et al. (2017, Front. Plant Sci.). For all other mutants for which no expression data was reported previously, we performed qRT-PCR in the revised manuscript to verify the expression level of the targeted gene in each mutant line. The results are presented in the new Supplementary Fig. 25. We furthermore followed the reviewer's suggestion and organized the relevant information for each mutant in the new Supplementary Table 5.

- 200 different accessions are used to determine natural variation of LR length to N availability but you only show 9 different ecotypes to conclude that there is no relationship between LR response and YUC8 expression. I understand that you have representative natural accessions with contrasting N responses, how many of these 9 ecotypes are representing mild, medium and high responses to low N?

Response: Among the selected accessions, Co, Ty-0 and Edi-0 are weak responders (increase of average lateral root length under low N relative to high N of less than 40%), Col-0 and Ven-1 intermediate responders (increases of 50% and 95%, respectively), while Kas-2, JEA, Tamm-27 and Uod-1 represent strong responders (increases of more than 100%).

Why not including Par-3, which present the highest increase in LR length?

Response: We did not include Par-3 as we did not have sufficient amounts of well-germinating seeds at the time when the experiment was carried out. However, we included Uod-1, an accession that showed a 185% increase in average lateral root length in low N compared to high N, which was very similar to the 188% increase shown by Par-3. For clarification, we now added the percent increase of lateral root elongation (LN-to-HN ratio) on top of the bars in the revised Supplementary Fig. 15a (Supplementary Fig. 10a of previous version of manuscript).

At least for Co, there can be a relationship between root response and YUC8 expression.

Response: We would like to point out that interpreting the data based on a single genotype is not meaningful to establish a relation for the observed intraspecific variation. Instead, *YUC8*-based association mapping and subsequent haplotype analysis were helpful in elucidating the genetic basis contributing to the natural variation. In a second step, we then compared *YUC8* expression in different natural accessions belonging to different haplogroups.

It is also not clear why the authors decided to search SNPs in 139 accessions and not the 200 initially used. The authors should explain this briefly.

Response: We did not apply any criterion to select 139 accessions for *YUC8*-based association analysis. In the GWAS analysis, we phenotyped 200 accessions that are genotyped with a 250K chip (Atwell et al., 2010, Nature; Horton et al., 2012, Nat. Genet.). However, of these 200 lines, complete genomic information is available for only 139 accessions which have been re-sequenced (1001

Genomes Project; <http://signal.salk.edu/atg1001/3.0/gebrowser.php>). This point is now clarified in the text (*Lines 411-414*).

- l.212: You conclude that “LN-induced LR elongation relies on BR signaling-dependent stimulation of YUC5/7/8 expression to increase local auxin levels”, however you showed that there was no relationship between YUC expression or change in expression in response to LN and root responses. How is then this conclusion supported?

Response: This conclusion is based on the results we obtained with BSK- and BZR1-dependent expression of *YUC8* and other *YUCs* (Fig. 5f, g; Supplementary Fig. 23 of the revised manuscript), the auxin reporter expression in the *bzr1-1D* background (Fig. 5h-k) and the chemical complementation of the *bsk* mutants by external IAA (Fig. 5e). The point made by the reviewer may refer specifically to the variation of the lateral root foraging response among natural accessions, where no correlation between *YUC8* expression and lateral root response to low N was observed. In this context, allelic variation in the coding sequence of *YUC8* is more relevant to explain phenotypic differences between accessions, as also confirmed by our transgenic allelic complementation experiments shown in Figs. 3 and 4.

- Is this YUC-dependent LR elongation phenotype dependent on a specific source of N, or is specifically connected to nitrate deficiency?

Response: As we did not intend to assess the effect of nitrate or ammonium but rather to decipher the mechanism underlying changes in lateral root elongation in response to external N levels previously determined to induce mild deficiency in shoots (Gruber et al., 2013 Plant Physiol.), we supplied N as NH_4NO_3 and KNO_3 and lowered the supplied concentration from 11.4 mM to 0.55 mM without altering the ratio between NH_4NO_3 and KNO_3 (i.e., 1:9.4 in both cases).

- It is widely reported that N-elicited root architecture modifications are mediated by nitrogen and phytohormone crosstalk. For instance, nitrate interplay with abscisic acid, auxin, and cytokinin has been reported to locally and systemically control root architecture. The authors did not discuss this topic in the introduction. The introduction needs to be improved by citing all relevant prior work.

Response: Thanks for pointing this out. We have now substantially revised the Introduction to refer to aspects that have been already reported by previous studies. *Lines 42-44 and 51-54*.

- T-Allele is mentioned to be associated with longer LR in response to LN. However, despite being statistically significant, the difference shown in Supplementary Figure 1a between A and T-Alleles under an LN regime is almost negligible.

Response: As root architectural traits including root length are quantitative traits under the control of multiple loci with minor to moderate effects, it is not surprising that the difference between two alleles is not very large. This has been exemplified in many studies (Lachowiec et al., 2015, Plos Genet.; Kobayashi et al., 2016, Plant Cell & Environ.; Tang et al., 2018, Nat. Commun.; Li et al., 2019, Nat. Commun.; Ogura et al., 2019, Cell; Jia et al., 2019, Nat. Commun.), where the detected loci contributed only 10% to 20% of the observed phenotypic variation for a given quantitative trait.

Authors should show representative images of accessions corresponding to A and T-alleles to evidence the identified phenotype's relevance.

Response: Thank you for this suggestion. We now included this information in revised Fig. 1a.

- In Figure 1d the authors state that they detected one prominent peak associated with the LR response to low nitrogen on chromosome 4. Nevertheless, it can be observed that another peak is present on chromosome 4 which was omitted by the authors. In this respect, the authors should clarify why they only studied one peak and not the other.

Response: Actually, we chose this SNP because 1) it is significantly associated with LR response, 2) is supported by adjacent markers, and 3) T-DNA insertion lines were available for all genes falling within the 20-kb supporting interval, allowing us to determine experimentally which of the genes sitting within this interval was the most promising candidate for further investigation. The outcome of our experiments enabled us to establish a significant impact of YUC8 on the investigated response. We followed the suggestion of Reviewer 1 by mentioning the other significant peak in *Lines 86-91*.

- The authors showed that the identified SNP was located in YUC8 gene. In this regard, they analyzed T-DNA insertion lines of YUC8 and two other genes located within the 20-kb interval centered around the identified SNP. The authors observed no significant statistical differences for average LR length between Col-0 and T-DNA insertional lines for At4g28730 and At4g28740. Nevertheless, they only evaluated one mutant allele per gene. The authors should also evaluate the response of other mutant alleles for the studied genes in order to have a more robust result. In addition, the authors should verify through real-time qRT-PCR that the T-DNA insertional mutants show a decrease on the gene's transcripts when compared to Col-0 plants.

Response: Due to limited time for revision, we were not able to isolate a second homozygous mutant allele for At4g28730 and At4g28740. However, we could verify by qRT-PCR that the T-DNA insertion lines for At4g28730 (SALK_077059C) and At4g28740 (SAIL_1286_E04C) represent real knock-outs (new Supplementary Fig. 25a,b). In addition, we found that although both genes are expressed in roots, none of them is up-regulated by N deficiency (Supplementary Fig. 1, new panels d and e), providing additional evidence that they unlikely contribute to root elongation under low N. Moreover, their functions are annotated as "iron-sulfur cluster assembly" and "chloroplast organization", which have so far no obvious relation to N-dependent root elongation.

The authors evaluated root architectural traits for YUC8 closest homologs in root: YUC3, YUC5, and YUC7. For this purpose, they used single mutants. Nevertheless, and as stated above, the authors should evaluate more than one mutant allele for each gene in order to have more robust results.

Response: In the revised manuscript, we confirmed by qRT-PCR that *yuc5-1* (SAIL_116_C01) and *yuc7-1* (SALK_059832C) represent real knockouts (new Supplementary Fig. 25d and f). In addition, we isolated the new knockout line *yuc5-2* (SALK_088618C) and *yuc7-2* (SALK_034074C). We phenotyped *yuc5-2* and *yuc7-2* under mild N deficiency and found that lengths of primary root as well as lateral roots were significantly shorter than those of wild-type Col-0 plants (new Supplementary Fig. 3 and 4), validating the phenotypes previously recorded for *yuc5-1* and *yuc7-1*.

In addition, the *yuc3* (GK-376G12) mutant has the T-DNA insertion in an intron. Thus, the authors must complement the root phenotypic analysis with a qRT-PCR in order to assess if the mutant lines actually show a decrease in the gene's transcript when compared to control plants.

Response: Isolation and validation of the root phenotype of *yuc3* with a second homozygous allele was not possible because of limited time for revision. Therefore, we decided to remove the *yuc3* dataset from the manuscript. This change does not affect any of the major messages of our manuscript.

- In Supplementary Fig.3, the authors observed that *yuc3*, *yuc5*, and *yuc7* mutants show a decrease in primary root and LR lengths under N deficiency. However, the *yuc5* mutant has a Col-3 background and the authors compare it with Col-0. Thus, the authors must compare the response of the *yuc5* mutant line with a Col-3 line.

Response: Thanks for spotting this. We repeated the experiment by comparing the root responses of *yuc5-1* (SAIL_116_C01) to mild low N with Col-3. We found that root elongation including primary root and lateral roots of *yuc5-1* was substantially decreased compared with wild type under low N. These results are now included in the Supplementary Fig. 3 and mentioned in the revised text (Lines 119-121).

- Supplementary Table 1 with information about the accessions used in this study is missing and Supplementary Table 2 containing information about SNPs in YUC8 coding sequence are missing. Supplementary Table 3 with information about the two major haplotypes is also lacking. Supplementary Table 4 with the primers used in this study is missing as well.

Response: From our perspective, it appears that there was a problem only for Reviewer 2 to access these Supp. Tables. We are sorry for this but confirm that the tables were and are still present.

- Line 149 is not finished. The position of the substitution must be stated.

Response: We apologize for this mistake. We now added the “position 14” in the revised version (Line 189).

- The authors should report the experiments by which they confirmed *yucQ* transformation with the YUC8-hap A and YUC8-hap B haplotypes.

Response: As stated in lines 206-210, we assessed the expression of YUC8 in several independent transformants by qRT-PCR and selected for the phenotypical analysis three independent T3 homozygous lines showing comparative expression for each construct. The results were presented in the Supplementary Fig. 12a of the previous and Supplementary Fig. 18a in the revised version of manuscript.

- The authors transformed *yucQ* mutants with a Col-0 promoter in Figure 3. However, they did not specify the *yucQ* background. If this mutant did not have a Col-0 background, how can the authors be sure that phenotypes observed in *yucQ* and *yucQ* complemented lines are comparable with Col-0? The *yucQ* background needs to be specified.

Response: The comparison of *yucQ* with Col-0 has been widely used in previous studies and therefore appeared to us as being accepted by the community (see e.g., Matosevich et al., Nat. Plants, 2020; Gaillochet et al., 2020, Development; Zhu et al., 2019, J Biol. Chem.; Paris et al., 2018, Front. Plant Sci.; Tsugafune et al., 2017, Sci. Report; Liu et al., 2016, Plos Genet.; Sugawara et al., 2015, Plant Cell Physiol.; Li et al., 2012, Genes & Dev.). Nonetheless, we tracked back the genetic background of the single mutants and found that it is Col-0 for YUC3, YUC7, YUC9 and most probably for YUC8 (assigned only as Col), while it is Landsberg for YUC5. Unfortunately, the original paper by Chen et al. (2012) does not provide information on the order of the crossings to estimate eventual background effects. However, we note that the main conclusions based on our *yucQ* experiments are not to show an additive effect of several YUCs to the root phenotype but rather to serve as a multiple *yuc* deletion line, in which we quantify the relative contribution of YUC8 haplotypes under low N versus sufficient N. Building our conclusion on this relative assessment makes the impact of the genetic background

less relevant, since it just determines the size of the biological effect. We now added this information in the revised manuscript, *lines 369-370*.

- In figure 3e-f, authors compared PR and LR length between mutant and complemented genotypes. The authors concluded that YUC8-hapA variant rescued more efficiently PR length. However, the numbers used in these experiments are not comparable. Despite agreeing with the mentioned conclusions, we suggest performing a statistical analysis with a comparable N number.

Response: To address the reviewer's concern, we now also performed a Welch's *t*-test to specifically compare the lines complemented with YUC8-hap A and YUC8-hap B (note that both had similar N numbers of 110 and 113, respectively). This additional statistical test provided further support that the YUC8-hapA variant is more efficient in rescuing PR and LR elongation of *yucQ*. The Figure 3e-f has been revised accordingly.

- In Figure 4 (a-c), the letters showing statistical differences are too big and the box plots too small, making it difficult to read the graphs. The same can be said for Figure 4 e-f; Uppercase letters should be placed next to the boxplot. The graphs in this Figure must be improved.

Response: Thanks. We now decreased the size of letters and reformatted the Fig. 4 in the revised manuscript.

The letters showing statistical differences in Fig. 5 b and c are too big when compared to the boxplots. The authors should improve the format of the graphs.

Response: Thanks. We now decreased the size of letters in the revised Fig. 5b-c.

- Most of the discussion is focused on explaining why the YUC8-Hap A is associated with LR growth under a low N concentration, and states the novelty of this study. However, the authors poorly contrast their results with the known signaling pathways, molecular components and already proposed models of N signaling underlying root responses. Are there any N- responsive transcription factors that could be candidates for the control of BR - associated genes?

Response: As suggested by the reviewer, we now also refer more extensively to other signaling pathways in the revised text (*Lines 147-158* in Results where we present the results of *AGL21*; and *Lines 350-353* in Discussion where we refer to CEP-CEPRs-CEPDs systemic N signaling pathway). However, in order to accommodate further points of discussion requested by other reviewers and avoid simple comparisons between pathways that respond to different N scenarios, we did not expand too much the discussion on known signaling pathways that are not specifically involved in the regulation of root foraging responses to mild N deficiency.

Are these results consistent with any other models of auxin - mediated control of root architecture in response to N availability? How can the study's results be incorporated into these already proposed models?

Response: Thus far, most known auxin-mediated control of root development (e.g., *miR167/ARF8* and *miR393/AFB3* modules regulating LR initiation and progression; and NRT1.1-dependent auxin transport in LR emergence) are restricted to responses to high nitrate or to severe N deficiency (Gifford et al., 2008; Vidal et al., 2010; Krouk et al., 2010; Bouguyon et al., 2015). As mentioned above, we also show in the revised version that *AGL21* and *ANR1* do not play a significant role in root growth responses to mild N deficiency and YUCCA-dependent local auxin biosynthesis is not required for root survival response to extremely low N availability. Therefore, our results and the findings of

previous study suggest that distinct signaling modules coordinate the characteristic changes induced by nitrate supply, or by moderate or severe N limitations. We now discuss these aspects more clearly in the revised manuscript (*Lines 147-158 and 278-281*).

- Line 66-67: Plants are not grown in high or low N for 9 days. Plants are grown for one week on sufficient N, plus 9 days on HN or LN. Please change to make it more clear to the reader.

Response: Agreed. We now amended the sentence to “After transferring 7-day-old seedlings precultured on sufficient N to HN or LN for 9 days” *Lines 79-80*.

- Fig 1c: This figure is not showing relative average LR length, as indicated in the legend. Please change “relative average LR length” for an x-axis title “LN/HN LR length ratio”

Response: Thanks, done!

- Line 82: “we nonetheless analyzed” I think this conjunction is not well used in this context. Maybe change to “we then analyzed...”

Response: Thanks, done!

- Supp Fig. 2ab: There are no boxes indicating what the colors mean.

Response: Thanks for spotting this mistake. We now include a legend on the top of the Supplementary Fig. 2a,b.

- Supp Fig. 2A should be included in Fig 1, since it is part of the root morphological analysis of the WT vs. YUC8 mutants. Total root length graph is redundant since there is already a plot showing PR length and LR length.

Response: Considering that lateral root number is not significantly different between WT and *yuc8* mutants and that Fig. 1 is already quite crowded, we prefer to leave these results in Supplementary Fig. 2.

- Figure 2: You have already shown a LR phenotype for the *yuc8* mutants, why center the cell elongation analysis on a mutant that includes genes that are not expressed in roots? The *yucQ* mutant is associated with an important decrease of IAA content that can mask a more-specific, YUC8-N-related phenotype.

Response: In Fig. 2a-d, our intention is to estimate the role of YUCCA-dependent auxin biosynthesis in increasing meristem size and cortical cell length to stimulate lateral root elongation.

- Line 112: “Conversely, when YUCCA-dependent auxin biosynthesis in roots of wild-type plants was suppressed with 4-phenoxyphenylboronic acid (PPBo)...” Since PPBo is an inhibitor of YUCCA activity, how can you be sure that auxin biosynthesis is specifically suppressed in the roots?

Response: As shown in Supplementary Fig. 7, to inhibit auxin biosynthesis in roots, we supplied PPBo to the root-containing agar while removing the agar on the top part containing the shoots.

- Supp Fig. 8. What happens with PR length response to low N? Is it also altered by NPA (I can't really tell from the photograph)? A graph of PR root length should also be added to Supp Fig 8 to support the conclusion that “these results collectively indicate that YUCCA-mediated local auxin production in roots modulates root elongation under mild N deficiency”

Response: Compared to mock, the supply of 5 μ M NPA to the shoot compartments slightly but significantly inhibits PR elongation under both low and high N. However, the response of PR length to low N (i.e., LN-to-HN ratio) was not altered by NPA supply to shoots, corroborating that polar auxin transport did not affect low N-induced root elongation. Following the reviewer's recommendation, we now include this dataset in revised Supplementary Fig. 10.

- This paragraph (l. 128-137) should be included before the analysis of gene expression, to support the conclusion that YUCCA-mediated local auxin production modulates root elongation under mild N deficiency (l.118-119).

Response. Thanks for this suggestion. However, we think that the increased root auxin accumulation revealed by the *R2D2* reporter is a consequence of enhanced expression of *YUC* genes under mild N deficiency. Following this logic, we prefer not to modify this paragraph.

- Supp Fig. 9: Please include panel a) on Fig 2e

Response: Thanks. GUS staining was performed at both 7 and 9 days after transfer to N treatments. To remain consistent with the LRs shown in Fig. 2e in previous and Fig. 2i in present version of manuscript, we moved stained PRs from plants grown for 9 days on the treatments to Fig. 2i.

Reviewer #3 (Remarks to the Author):

In this manuscript, Jia et al. takes advantage of different Arabidopsis accessions carrying natural variation to evaluate their root response to nitrogen deficiency, intending to seek the components involved in this biological process. Using GWAS and SNP analysis, the authors identified a target gene YUC8, an auxin biosynthesis-related gene, and proposed that this gene's family members contribute to lateral root elongation response to N deficiency. The genetic experiments using *yuc* single or multi-mutants further confirmed that the YUC-mediated local auxin biosynthesis process in root is associated with the root response to nitrogen. Consistently, the YUC genes' expression levels could be triggered by N deficiency. Meanwhile, the authors identified two allelic variants of YUC8 harboring a non-synonymous substitution of one amino acid. They demonstrated that the different biological activity between YUC8 variants underlies the differential lateral root foraging for nitrogen for these Arabidopsis accessions. Moreover, the genetic combined with the pharmacological experiments imply the auxin signaling pathway is downstream of the BR signaling in the root foraging for nitrogen. Generally, this work seems interesting, especially the crosstalk signaling between plant hormones brassinosteroid and auxin in root foraging for N. The data are also well organized and the paper is very easy to be followed.

Response: Thank you for the positive comments on our work.

However, some of the conclusions drawn from the data are not so persuasive, and I suggest the authors revise the text or add much more data to make it more explicit and accurate. Meanwhile, some analyses are a bit superficial. I have some comments below for authors to improve.

Major comment

1. The information conveyed by this title is not exactly what the data can support. The GWAS analysis results only conclude that the allelic variants of YUC8 contribute to the differential local auxin biosynthesis activities among these Arabidopsis accessions and thus trigger their different root foraging for nitrogen, but it cannot be used as the regulation mechanism in plants as title proposed. Concerning the mechanism, from the existing data, I believe that the auxin biosynthesis triggers root

foraging for nitrogen, but no data here shows that only YUC genes are up-regulated in the mild N deficiency. Are the expression levels of other crucial auxin biosynthesis-related genes (e.g. TAA) up-regulated under nitrogen deficiency? Do their corresponding chemical inhibitors (e.g. L-kyn inhibitors TAA activity) also disturb the lateral root elongation in low nitrogen as YUC inhibitor PPBo shows? If yes, it should be very careful to use the “YUC-dependent local auxin biosynthesis”, because other auxin synthesis genes are also involved in this process. Therefore, more experiments should be considered to enhance the view as the title proposed or modify the title.

Response: Thank you for these constructive suggestions. According to Ma et al. (2013, Plant J), both *TAA1* and *TAR2* are induced by mild N deficiency in the roots. In the same study it was also shown that *TAR2* is responsible for LR emergence but does not affect N-dependent LR elongation, probably because expression of *TAR2* is restricted to the root maturation zone. Instead, *TAA1* is expressed in the vasculature and root apical meristem, which overlaps with expression domain of *YUC8* and its homologous genes (Yang et al., 2014, Plant Cell; Chen et al. 2014, Plant Cell Physiol.; Brumos et al., 2019, Dev. Cell; Brumos et al., 2020, Plant Cell). In the revised version, we confirmed that *TAA1* is up-regulated by mild N deficiency and demonstrate that it is also under the control of the BR signaling transcription factor *BZR1* (new Supplementary Fig. 8). Furthermore, we assessed the root phenotypes of two *TAA1* mutant alleles (*ckrc1-1* and *wei8-1*) and found that they are also unable to stimulate root elongation in response to mild N (new Supplementary Fig. 9), suggesting a major role for *TAA1* together with *YUC5/7/8* in this process. Taken together, these results suggested that local auxin biosynthesis produced by *TAA1*-*YUC5/7/8* module acts downstream of BR signaling in stimulating root elongation in response to mild low N. We consequently amended our title to “Local auxin biosynthesis acts downstream of brassinosteroids to trigger root foraging for nitrogen”.

2. Usually, the genetic variations of geographically representative Arabidopsis accessions confer their fitness in a certain habitat. However, I did not see any this kind of analysis throughout the paper. To go much deeper, I urge the authors to further check the geographical distribution between the two groups of Arabidopsis accessions carrying the YUC8 variant YUC8-hapA and YUC8-hapB separately. Then combined with the feature of the local environments, it might provide some insights regarding how the auxin-dependent root foraging for N is modified by natural selection and linked to their adaptability to local environments. At least, it should be discussed in the discussion section.

Response: Thanks for this interesting suggestion. We exploited data of a previous climate adaptation study (Hancock et al., 2011) and observed that there were indeed significant positive correlations between the SNP_Chr4_14192732 and temperature seasonality and minimum temperature (new Supplementary Fig. 24a,b). Furthermore, we detected a clear association between the distribution of YUC8 allelic variants and temperature-related variables. As we now show in new Supplementary Fig. 24c-f, accessions harboring the YUC8-L allele grow on habitats with higher mean diurnal range, temperature seasonality, temperature annual range and mean temperature of wettest quarter as compared to accessions with the YUC8-S variant. These observations may suggest that *YUC8* variants play an adaptive role under prevailing temperature variability and support the role of YUC8-mediated auxin biosynthesis in plant adaptation to temperature variation (Sun et al., 2014, Plos Genet.; Lee et al., 2014, Nat. Commun.; Fiorucci et al., 2019, New Phytol.; Bellstaedt et al., 2019, Plant Physiol.). These findings are now presented and discussed in *Lines 300-312*.

3. Performing experimental analysis with only Arabidopsis accession Col-0 to verify the GWAS results is not so convincing. If some other Arabidopsis accessions (e.g. Co) can be used for similar experimental analysis, the conclusions would be much more solid. For example, even though the

bioinformatic analysis (GWAS, SNP) proposed that YUC8 coding variants (hapA and hapB) link to the diverse root response to N among these Arabidopsis accessions. However, no further data demonstrate if the differential transcriptional level of YUC8 is also involved in the diverse nitrogen responses or not.

Response: In order to assess the possible involvement of differential transcript levels of *YUC8* in root responses to low N, we determined transcript levels of *YUC8* in 9 natural accessions showing quantitative variation in their lateral root lengths and N responses. Of these 9 accessions, Co, Ty-0, Edi-0 and Tamm-27 express the *YUC8-hap B* variant, while Col-0, Ven-1, Kas-2, JEA and Uod-1 express the *YUC8-hap A* variant. According to our results i) there was no significant correlation between lateral root length and transcript levels of *YUC8* at either N condition; ii) no significant correlation was detected between fold-change (LN-to-HN ratio) of *YUC8* expression and lateral root length ratio; and iii) we found no difference in *YUC8* expression between two *YUC8* haplogroups at either N condition. We presented these results in Supplementary Fig. 10 in the previous manuscript and Supplementary Fig. 15 in the revised manuscript. Together, these results suggest that in the investigated lines *YUC8*-dependent natural variation under LN is unlikely due to variations at the transcript level.

Even though to request some transgenic work is not so reasonable, some other simple experiments can be performed using other Arabidopsis accessions. Is the transcription level of YUC8-hapB in accession Co also up-regulated under low N condition? Is its increased expression level of YUC8-hapB comparable to that of YUC8-hapA in Col background?

Response: i) We found that *YUC8* transcript levels in the accession Co increased by 17%, which is not significantly different between the two N conditions. ii) In Col-0, *YUC8* increased by 41%, which was not much different to Co. Furthermore, we selected lines with comparable expression for the allelic complementation experiment (Supplementary Fig. 18).

Moreover, to further prove that the different auxin accumulations of these Arabidopsis accessions lead to their varied root response to nitrogen, the exogenous 50nM IAA can be applied in different Arabidopsis accessions, to see if the elimination of endogenous auxin accumulation differences would abolish the differences of root N response among these Arabidopsis accessions.

Response: Thanks for this suggestion. Although external application of auxin to YUC8-hap A and -hap B accessions is expected to abolish the differences in endogenous auxin accumulation, the interpretation of the results will be complicated by genotypic differences in auxin uptake, transport and signaling. Therefore, we decided to supply the YUCCA inhibitor PPBo (5 μ M) to roots of 8 accessions from YUC8-hap A (Uod-1, Or-0, Wt-5, Kas-2, JEA, Ven-1, Pog-0, Ri-0) and 6 accessions from YUC8-hap B (Co, Edi-0, Tha-1, Ty-0, Alst-1, Tamm-27). Whereas under mock treatment the root foraging responses (i.e. LN-to-HN ratio of PR, LR and total root length) of YUC8-hap A accessions were on average significantly stronger than those of hap B accessions, exogenous supply of PPBo strongly abolished these differences. These results are now presented in the new Supplementary Fig. 20 and mentioned in *Lines 217-223*.

4. Something needs to be further clarified. The author assessed 200 Arabidopsis accessions and then identify YUC8 is involved in the diverse LR response to low N among these accessions. However, to identify the SNPs within YUC8 coding sequence, they used 139 Arabidopsis accessions. It is better to explain why different numbers of accessions are used for these two analyses and what criteria they applied for selecting the accessions to perform these bioinformatic analyses.

Response: We apologize for not clearly explaining this point in the first version. We did not apply any criterion to select 139 accessions for YUC8-based association analysis. In the GWAS analysis, we phenotyped 200 accessions that are genotyped with a 250k chip. However, of these 200 lines, complete genomic information is available for only 139 accessions, which have been re-sequenced (1001 Genomes Project; <http://signal.salk.edu/atg1001/3.0/gebrowser.php>). This point is now mentioned in the Methods. *Lines 411-414.*

Minor comment

1) Some descriptions are a bit overstated in the Abstract section. Page 1 line 11, it should be the “flowering plants”, not the “plants”.

Response: Thanks, we changed to “flowering plants”.

Line 32, this work only implies the ‘root response to nitrogen availability’ but not the ‘root response to abiotic stress’.

Response: Thanks, we changed to “nitrogen availability”.

2) The concentration of these chemicals used for plant treatment should be clearly showed. It is better to mention them when they appeared in the main text for the first time. For example, it is very hard for me to find which concentration of NPA they used in both the main text and figure legends.

Response: Thanks for pointing this out. We now specified this information in the text and figure legends.

3) Line 229, it is hard to have the conclusion that “mild N deficiency stimulated local auxin biosynthesis in the root apical meristem”. Firstly, no evidence shows that the auxin biosynthesis genes (e.g. YUC) are only up-regulated in root apical meristem. Secondly, even though the R2D2 signal is enhanced in RAM, however, it is also possible that the excessive synthesized auxin in other tissue is transported to RAM. It is much accurate to say the “local auxin accumulation in the root meristem”

Response: Thank you for this suggestion. We now say “local auxin accumulation”.

4) Line 274, there are few minor grammar mistakes, check through the manuscript to correct them. e.g. Line 274, ‘were’ should be changed to ‘was’. Line 149, some words seem to be missing after ‘at’.

Response: Thanks. Done!

5) Some words are very hard to be digested. ‘Homozygosity’ should be replaced by ‘homozygote’.

Response: Thanks. Done!

Reviewer #4 (Remarks to the Author):

The manuscript of Jai et al describes the natural genetic variation in YUC8 is involved in auxin biosynthesis as cause for the root foraging response to low N. Overall, manuscript is well written, authors have done a great job by doing many detailed and in-depth experiments to support their findings and conclusions.

Response: Thanks for the encouraging comments about our work.

However, I have a number of questions/comments regarding the data presented and there are still some issues that authors should take into account.

1) Authors only focused on one GWA peak located on chromosome 4 where top marker SNP located close to YUCCA8 gene. However, GWA analysis clearly also identified other regions with significant association (on Chr. 4 and Chr. 2).

Response: The SNP on the Chr2 is not significant when a False Discovery Rate at $q = 0.05$ is applied. As such, we detected two significant SNPs on Chr 4 (SNP_2724898 and SNP_14192732). Among these, we focused on the SNP Chr4_14192732, as the corresponding peak was supported by adjacent markers and T-DNA insertion lines were available for all genes falling within a 20-kb supporting interval. We clarified this point in the revised text. *Lines 86-91.*

Even though further experiments carried out meticulously by authors to show YUC8 as determinant for LR response to low N. In order rule out bias approach to choose YUC8 region for follow up experiments, authors should also investigate whether other genes located close to top SNPs from other two significant associations (shown in Fig. 1d.) are contributing to the LR response to LN or not. Authors need to perform expression analysis under HN/LN and phenotyping using T-DNA insertion lines of the genes falls within 10 to 20 kb regions from top SNPs present on Chr. 2 and Chr. 4.

Response: Indeed, there were two further genes (At4g28730 and At4g28740) falling within a 20-kb supporting interval of the investigated SNP_Chr4_14192732. We investigated T-DNA insertion lines of both genes and observed that LN-induced LR length was similar to wild-type plants, and the expression of either gene did not respond to LN (Supplementary Fig. 1b-e), excluding an eventual role of At4g28730 and At4g28740 in regulating LR elongation under mild N deficiency. We did not consider genes falling within a similar interval of the other SNPs, because the outcome of our experiments revealed a significant impact of YUC8. Thus, our discovery of relevant YUC8 variation and its contribution downstream of BRs remains valid irrespective of a putative contribution of genes in another locus. We believe that unravelling the contribution of the other SNP is matter for another story and beyond the scope of this manuscript. To clarify our decision to focus on SNP_Chr4_14192732, we improved the description in *Lines 89-102.*

2) Results section is very well written but introduction and discussion sections are too short. Authors should rewrite both the sections. Authors can further enrich the introduction part by giving better background on the LR development and N availability. For example: adding “role of TAR2 in low nitrogen-mediated reprogramming of root architecture in Arabidopsis” etc etc.

Response: Thanks for this suggestion. We now substantially expanded the Introduction and Discussion.

3) Authors should also discuss Auxin-BR crosstalk in more detail instead of writing just few lines on this interesting aspect. Also, in this case authors can prepare a schematic illustration describing the crosstalk between auxin-BR signaling and its role in LR growth under LN.

Response: Thanks for this suggestion. We revisited and substantially revised this aspect and now provide a schematic illustration in the revised manuscript (new Figure 6).

4) Figure 4 seems very complex, I would strongly suggest authors to replot this figure in a different way for better understanding for the readers.

Response: Thanks. We now reformatted Fig. 4 in the revised manuscript.

5) In Methodology section sufficient information is not given, for example haplotype analysis part is too short.

Response: We apologize for not being sufficiently clear. We performed *YUC8* gene-based association analysis by using identified SNPs from 139 natural accessions whose *YUC8* sequence is available to download from the 1001 Genomes Project (<http://signal.salk.edu/atg1001/3.0/gebrowser.php>) and have phenotypic data in our initial screening. The 6 resulting significant SNPs were taken to classify *YUC8* haplogroups. We only considered those haplotypes, for which more than five accessions were available for further comparative analysis. This information has now been included in the revised Methods. *Lines 411-420.*

Also, in many experiments sample size varies a lot and in some cases number of individuals are not optimum in my opinion. It is important to mention in that case whether experiments have been repeated multiple times or not.

Response: In Fig. 3e-g, Supplementary Fig. 11 and Supplementary Fig. 12 c-e in the previous version (Fig. 3e-g, Supplementary Fig. 17 and Supplementary Fig. 18 c-e of revised version), we pooled all 6, 20 and 3 independent transgenic lines expressing the same construct. Following the reviewer's recommendation, we now clarified these points in the corresponding legends. Furthermore, we repeated all experiments at least once and presented the results from one representative experiment.

6) Authors last paragraph (Line 267-270) in discussion section is superficial and overclaiming.

Response: We deleted this paragraph and added instead the following sentence: "Under any of these constraints, employing CRISPR-mediated gene editing to turn "weak" *YUC8* variants into "strong" variants could provide an opportunity to increase root elongation and subsequent water and nutrient acquisition in crops". *Lines 355-358.*

REVIEWERS' COMMENTS

Reviewer #1 (Remarks to the Author):

I would like to say well done to the authors in pursuing the improvements of their manuscript and am glad that my comments were useful. In this revision the authors explain the rationale for the choice of the 200 accessions and discussion of hap A vs. hap B accessions; I am sure that this will continue to be an area of interest as we learn more about more subtle molecular differences amongst accessions in the future. In addition, they help better place the work in context of previous physiological-focussed BR-auxin-N interactions, addressing the major comment of reviewer #2, explained the choice of significant SNP region (all reviewers), further elaborated on the evaluation of *YUC8* expression levels (reviewer #3) and developed the detail in the text (reviewer #4). All of these have improved the manuscript and I enjoyed the discussion provided in the response to reviewers.

I am sure that the manuscript will be of broad interest to both plant scientists and any researchers interested in how molecular signalling is integrating at the local vs. systemic level to control a phenotypic response.

Response: We thank the Reviewer for the positive comments.

Reviewer #2 (Remarks to the Author):

Thanks to the author for constructively addressing my comments. This version of the manuscript satisfies my concerns.

Response: We thank the Reviewer for the positive comments.

Reviewer #3 (Remarks to the Author):

The author(s) have addressed all my concerns, and generally, the revised manuscript is greatly improved. In this version, the author(s) analyzed the other auxin biosynthesis genes and indicated that the local auxin biosynthesis is involved in the root foraging for nitrogen. They further evaluated the geographic distribution of the two groups of Arabidopsis accessions separately carrying YUC8-hapA and YUC8-hapB, suggesting that the genetic variations link to the plant adaptability to temperature differences. The authors also give detailed information regarding the Arabidopsis accessions used for the phenotypic and GWAS analysis. Meanwhile, the conclusion is much more solid and the interpretation is more precise after the revision.

Response: We thank the Reviewer for the encouraging comments.

I only have some very minor points irrelevant to the academic question, which should be considered by the author for improvement.

1. In Abstract (Line 19), please give the full name of the 'TAA'.

Response: Thanks, done!

2. In Supplementary Fig 8 and Supplementary Fig. 23, for these statistical analysis, the symbols 'ns', '*' and '**' above the bars confused me a lot, because they were everywhere. I am not so sure which data sets were used for comparison to see the significant differences. Author should clarify it in relevant figure legends to make it clear.

Response: To make it be easily understood, in the final version of manuscript we colored the P-value for pairwise comparison between genotypes and N treatments.

Reviewer #4 (Remarks to the Author):

It appears to me that most of my concerns have been appropriately addressed by authors and the manuscript has been greatly improved.

Response: We thank the Reviewer for the positive comments.